

# Numerical study of the seasonal thermal and gas regimes of the large artificial reservoir in Western Europe using LAKE2.0 model

Maksim Iakunin[1], Victor Stepanenko[2], Rui Salgado[1], Miguel Potes[1], Alexandra Penha[3,4], Maria Helena Novais[3,4], and Gonçalo Rodrigues[1]

[1]Institute of Earth Sciences — ICT, University of Évora, Department of Physics, Rua Romão Ramalho 59, 7000-671 Évora, Portugal
[2]Lomonosov Moscow State University, GSP-1, 119234, Leninskie Gory, 1, bld. 4, Moscow, Russia
[3]Water Laboratory, University of Évora, P.I.T.E. Rua da Barba Rala Nº1, 7005-345 Évora, Portugal
[4]Institute of Earth Sciences — ICT, University of Évora, Rua Romão Ramalho 59, 7000-671 Évora, Portugal

**Correspondence:** Maksim Iakunin (miakunin@uevora.pt)

**Abstract.**

The Alqueva reservoir (southeast of Portugal) being the largest artificial lake in Western Europe and strategic freshwater supply in the region is of scientific interest in terms of monitoring and maintaining the quality and quantity of water and its impact on the regional climate. To solve these tasks we conducted numerical studies of the thermal and gas regimes in the lake over the period from May 2017 to March 2019, supplemented by the data observed at the weather stations and the floating platforms during the field campaign of the ALOP (ALentejo Observation and Prediction System) project. One-dimensional model LAKE2.0 was used for the numerical studies. Being highly versatile and adjusted to the specific features of the reservoir, this parameterization is capable to simulate its thermodynamic and biogeochemical characteristics. Profiles and time series of water temperature, sensible and latent heat fluxes, concentrations of $CO_2$ and $O_2$ reproduced by the LAKE2.0 model were validated against the observed data and were compared with the thermodynamic simulation results obtained with the FLake model. The results demonstrated that LAKE2.0 model has good ability in capturing the seasonal variations in the water surface temperature and the internal thermal structure of the Alqueva reservoir, and satisfactorily captured the seasonal gas regime.

## 1 Introduction

Inland water bodies are active and simultaneously sensitive participants of the weather and climate processes of the Earth, changing the temperature, wind, precipitation in the surrounding areas; their thermal and gas regimes, in turn, can serve as a response to ecosystem status or climate change (Bonan, 1995; Adrian et al., 2009; Samuelsson et al., 2006). In modern climate/weather models lakes and reservoirs became large-scale structures and are taken into account explicitly (Bonan, 1995), their parameterizations are intensively embedding in these models (Mironov et al., 2010; Salgado and Le Moigne, 2010; Dutra et al., 2010; Subin et al., 2012). One-dimensional lake models play a major role in this process, e.g. the FLake model (Mironov et al., 2010). Their simplicity, computational efficiency and reliability of the simulation results allow to use them not only in studies of the dynamics of single lakes but also in the climate-related tasks of long-term numerical simulations,





where vast territories with huge number of water bodies should be taken into account. As a result, the number of numerical studies connected with the vertical thermodynamics and biogeochemistry of lakes and their interaction with the atmosphere increases (Thiery et al., 2014; Heiskanen et al., 2015; Le Moigne et al., 2016; Ekhtiari et al., 2017; Su et al., 2019).

A realistic representation of the thermal and gas regimes by lake models are important for solving current and prognostic tasks. For example, high accuracy of the calculations of sensible and latent heat fluxes, momentum, and water surface temperature is required for atmospheric models, where these parameters are the boundary conditions (Bonan, 1995; Mironov et al., 2010; Dutra et al., 2010; Salgado and Le Moigne, 2010; Balsamo, 2013). On the other hand, an adequate simulation of the water temperature profiles would be a very interesting new output of weather prediction and earth system models because the

temperature is a key factor for the lake ecosystem vital activity. This information might be useful for water quality management and for better representation of the gas emissions ($CO_2$, $O_2$, $CH_4$) from lakes to the atmosphere which are relevant to various atmospheric processes (Walter et al., 2007).

    Fully filled only in 2004, the Alqueva reservoir is in the spotlight of many studies connected with its ecosystem vital activity and ecology (Penha et al., 2016; Tomaz et al., 2017; Pereira et al., 2019), water quality (Potes et al., 2012, 2018; Novais et al.,

2018), and lake-atmosphere interactions (Lopes et al., 2016; Policarpo et al., 2017; Potes et al., 2017; Iakunin et al., 2018). The aim of the present work is a numerical study of the seasonal variations of thermal and gas regimes of the reservoir which was held under the ALOP (ALentejo Observation and Predicition systems) project, where an extensive field campaign and lake model simulations were combined. For the latter we used one-dimensional model LAKE2.0 (Stepanenko et al., 2016), that features the biogeochemical block allowing to reproduce the concentrations of $O_2$, $CO_2$, and $CH_4$ in water. In addition,

well-established in weather and climate studies FLake model was used as a reference to complete the results of thermodynamic characteristics of the reservoir. Before starting the numerical simulations, the LAKE2.0 model has been adapted to the features of the Alqueva reservoir including the introduction of the realistic values of the water acidity and light extinction coefficients and adequate value of the coefficient of the hipolimnion turbulent mixing rate. Both models were forced with the observed data at the reservoir which contributed to increase the reliability of the results. The simulation covered the period from May 2017

to April 2019 and its results as well as the possibility to apply LAKE2.0 model in operational mode might be used in future studies of weather and climate, and biochemical related tasks.

## 2   Methods

### 2.1   Object of study

The Alqueva reservoir is located in the southeast of Portugal spreading along 83 km over the former valley of Guadiana

river (Fig 1). Established in 2002 to cover the region's water and electricity needs, its surface covers an area of 250 $km^2$, the maximum depth is 92 m, the average depth is 16.6 m, and the capacity of water is estimated at 4.15 $km^3$, which make it the largest reservoir in Western Europe.

    Long periods of drought that could last for more than one consecutive year (Silva et al., 2014) are typical in this part of the Iberian Peninsula. The Alqueva region is characterized by a hot-summer Mediterranean climate (*Csa* type according to the





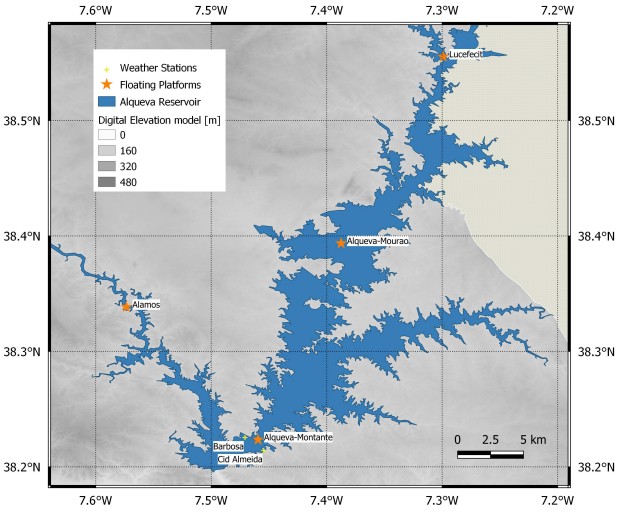

**Figure 1.** Location of the Alqueva reservoir and ALOP stations. The map was built using digital elevation model ASTER GDEM 2 (https://asterweb.jpl.nasa.gov/gdem.asp).

Köppen climate classification) with a small area of semi-arid climate (*BSk* type). In summer, maximum daily air temperature ranges between 31 and 35 °C (July and August) while the record values may reach 44 °C. Winter period (December-February) in the region is relatively mild and wet with average air temperature of 10.3 °C. Nevertheless, even in January the air temperature can reach maximum value of 24 °C during long periods of stable conditions when the Azores Anticyclone settles in favourable position. Rainfall seasons normally last from October to May. The annual average values of the accumulated

precipitation (1981-2010 normals from www.ipma.pt) registered at the weather station in Beja located 40 km away from the reservoir is 558 mm. Mean daily values of the incident solar radiation at the surface are about 300 $\mathrm{Wm}^{-2}$ (one of the highest in Europe) and the daily maximum in summer often may exceed 1000 $\mathrm{Wm}^{-2}$ (Iakunin et al., 2018).

## 2.2   Observed data

Geographical and climatological factors make the Alqueva reservoir vital source of fresh water for living and economical

purposes in the region. On the other hand, an increasing anthropogenic and heat stress negatively affects the lake ecosystem (Penha et al., 2016). Monitoring the quantity and quality of water in the reservoir became an essential scientific task. This task is addressed in the framework of the ALOP project related to the observations and numerical experiments on the study of processes of the atmosphere - Alqueva reservoir system. Models of different spatial and time scales were used in the ALOP numerical experiments.

The ALOP field campaign was focused on measurements of physical, chemical, and biological parameters in the water and air columns, over the water-atmosphere interface, and in the shores of the reservoir. In present work the following facilities were used and equipped to obtain the required data for the numerical simulations during the field campaign: 4 floating platforms





(Montante, Mourão, Alamos, and Lucetfecit) and two dedicated weather stations in the margins (Barbosa and Cid Almeida), their locations are marked with circles in Fig 1. The principal scientific site on the lake was the Montante floating platform

which is located in the southern and deeper part (74 m) of the reservoir (38.2276° N, 7.4708° W). The following equipment was settled on the platform during the whole field campaign continuously providing measurements:

– an eddy-covariance system, Campbell Scientific Irgason, provides data for atmospheric pressure, air temperature, water vapour and carbon dioxide concentrations, 3D wind components, linear momentum, sensible heat, latent heat, and carbon dioxide fluxes;

– albedometer (Kipp & Zonen CM7B) and pyrradiometer (Philipp Schenk 8111) in order to measure upwelling and down-welling shortwave and total radiative fluxes;

– set of 14 probes (Campbell Scientific 107) to measure the water temperature profile at the following depths: 5 cm, 25 cm, 50 cm, 1 m, 2 m, 4 m, 6 m, 8 m, 10 m, 12 m, 15 m, 20 m, 30 m, and 60 m.

Two probes were installed at the platform to assess water quality. A multiparametric probe (Aqua TROLL 600, IN-SITU,

USA) that provided information about dissolved oxygen concentration and pH values, among other parameters, was mounted on the platform at 25 cm depth on the 3rd of July 2018 and worked until the end of the campaign. It was also used to make profiles during regular maintenance visits to the platform. A Pro-Oceanus Mini CO2 Analog Output probe was also mounted on the platform at 25 cm depth to measure dissolved $CO_2$ concentration continuously and for punctually vertical profiles. Installed in the beginning of the campaign, the probe was working until the middle of June 2017 when it failed. It has been

repaired and installed again in October 2017 but another problem occurred in November and probe was removed definitely.

Two land weather stations (Barbosa and Cid Almeida) were installed on opposite shores with the floating platform in the middle, between them (38.2235° N, 7.4595° W and 38.2164° N, 7.4545° W, correspondingly, green circles in Fig. 1). The equipment of both weather stations is listed in Table 1. Data from the Montante floating platform, Barbosa, and Cid Almeida weather stations were obtained in automatic regime and transferred daily to the server in the Institute of Earth Sciences (ICT),

University of Évora. An important part of the campaign were the regular field trips to the reservoir for the cleaning and maintenance of the instrumentation on the platforms and weather stations and conduct measurements, to collect water samples at several depths and bottom sediments.

For further work, the data collected during the field campaign was treated before used as a forcing for atmospheric and/or lake modelling related tasks. Gaps, errors, and missed data were carefully interpolated.

## 2.3  LAKE2.0 model

For the simulation of the thermodynamic and biogeochemical processes in the Alqueva reservoir the LAKE2.0[1] model was chosen. A detailed description of the LAKE2.0 model may be found in Stepanenko et al. (2016), briefly the model equations are formulated in terms of water properties averaged over a lake's horizontal cross-section, thus introducing into the model fluxes

---

[1] Available at http://tesla.parallel.ru/Viktor/LAKE/wikis/LAKE-model





**Table 1.** Weather stations equipment

| Measured | Station | |
| --- | --- | --- |
| parameter | Barbosa | Cid Almeida |
| Albedometer | N/A | Philipp Schenk 8104 |
| Air temperature and humidity | Campbell Scientific CS 215 | Thies Clima 1.1005.51.512 |
| Wind Speed | Gill Instruments WindSonic 1405-PK-021 | Vector Instruments A100R |
| Wind Direction | ,, | Vector Instruments W200P |

of momentum, heat and dissolved gases through a sloping bottom surface. Water temperature profile is simulated explicitly in

LAKE2.0, and unlike Hostetler model (Hostetler and Bartlein, 1990), a number of biogeochemical processes is represented, which makes it capable to reproduce the transfer of $CO_2$ and $CH_4$ from and to the atmosphere.

Governing equations for the basic processes of the lake dynamics in the model are obtained using horizontally averaged Reynolds advection-diffusion equation for the quantity $f$ which may be one of the velocity components, temperature, turbulent kinetic energy (TKE), TKE dissipation, or gas concentration:

$$c\partial_t \bar{f} = \underbrace{A^{-1}\partial_z(Ak_f\partial_z\bar{f})}_{I} - \underbrace{A^{-1}\partial_z(A\overline{F_{nz}})}_{II} + \underbrace{R_f(\bar{f},\ldots)}_{III}, \tag{1}$$

where term $I$ describes turbulent diffusion, thermal conductivity or viscosity, term $II$ is the divergence of non-turbulent flux of $f$, term $III$ represents the horizontally averaged sum of sources and sinks, $\bar{F}_{nz}$ is the non-turbulent flux of $f$, $k_f$ is the turbulent diffusion coefficient (thermal conductivity coefficient for temperature, viscosity for momentum) for $f$ quantity. The LAKE2.0 model successfully represents conditions in the well-mixed upper layer of lakes (epilimnion).

In water, $k-\epsilon$ parameterization for computing turbulent fluxes is used. In ice and snow, a coupled transport of heat and liquid water is reproduced (Stepanenko et al., 2019). In bottom sediments, vertical transport of heat is implemented in a number of sediment columns, originating from different depths.

Water temperature profile in the model is driven by equation (1) with substitution $f \to T$, where $c = c_w\rho_{w0}$, $c_w$ — water specific heat, $\rho_{w0}$ — mean water density, $\overline{R_f} = 0$ represents heat flux from the sediments, $\overline{F_{nz}}(z) = S_{rad}$ — downward short-

wave radiation flux attenuating according to Beer–Lambert law in four wavebands (infrared, near-infrared, photosynthetically active, ultraviolet) with prescribed extinction coefficients. Heat conductance is a sum of molecular and turbulent coefficients, $k_T = \lambda_m + \lambda_t$, where $\lambda_t = c_w\rho_{w0}\nu_T$ ($\nu_T$ — turbulent coefficient of thermal diffusivity, $\text{m}^2\text{s}^{-1}$ derived from the $k-\varepsilon$ parameterization).

To solve the equation (1) for water temperature, top and bottom boundary conditions should be defined. The top boundary

condition are represented by a heat balance equation, involving net radiation and a scheme for turbulent heat fluxes in the surface atmospheric layer based on Monin–Obukhov similarity theory (Monin and Obukhov, 1954). Bottom boundary condition is set at the water-sediments interface and is based on the continuity of both heat flux and temperature at the interface. Bottom





sediments are represented with one-dimensional multilayer model which includes heat conductivity, liquid moisture transport (diffusion and gravitational percolation), ice content, and phase transitions of water.

Lake hydrodynamics is described by (1) applied to horizontal momentum components with $F_{nz} = 0$, $c = 1$, and $R_f$ representing Coriolis force and bottom friction. The Coriolis force has to be included in the momentum equations for lakes with horizontal size that exceeds the internal Rossby deformation radius (Patterson et al., 1984).

Wind stress which is computed by Monin-Obukhov similarity theory is applied as a top boundary condition for momentum equations, bottom friction is set by logarithmic law with prescribed roughness length. Friction at a sloping bottom (term $R_f$)

is calculated by quadratic law with tunable drag coefficient.

LAKE2.0 model uses $k - \varepsilon$ model (Canuto et al., 2001) to compute turbulent viscosity, temperature conductivity and diffusivity. It takes into account both shear and buoyancy production of turbulent kinetic energy; an equation for dissipation rate is a highly parametrized one with several constants calibrated in idealized flows.

Biochemical oxygen demand (BOD) is caused by degradation of dissolved organic carbon (DOC) and dead particulate or-

ganic carbon (POCD). The dynamics of the latter two, together with living particulate organic carbon (POCL) is represented by the model from Hanson et al. (2004) adapted to the 1D framework. Photosynthesis is given by Haldane kinetics where chlorophyll-a concentration is assumed to be constant in the mixed layer, assumed zero below and computed from photosynthetic radiation extinction coefficient (Stefan and Fang, 1994), an external model parameter. The model does not take into account the nutrients concentrations explicitly. The fluxes of dissolved gases to the atmosphere are calculated using Henry's law

and surface-renewal model (Stepanenko et al., 2016), involving subsurface turbulent kinetic energy dissipation rate, provided by the $k - \epsilon$ closure.

To calculate the dissolved carbon dioxide concentration in a water same type of prognostic equation is used as that for other gases. In LAKE2.0, sedimentary oxygen demand and BOD, respiration, and $CH_4$ oxidation act as $CO_2$ producers, while photosynthesis is the only sink of carbon dioxide in the water column.

## 2.4 Model modifications and sensitivity tests

The given version of the LAKE2.0 model used a constant light extinction coefficient in water. This could lead to significant errors, especially in long term simulations, because this parameter controls the vertical distribution of solar energy in different water layers and has a big annual variability in the Alqueva reservoir, as shown in Potes et al. (2012). On the other hand, light extinction coefficient was constantly measured during the ALOP field campaign. Since April 2017 until March 2019

it varied from a minimum of 0.247 (August 2017) to a maximum of 1.519 (July 2018) with an average value of 0.643 (12 measurements). Thus, prior to the simulation it was decided to upgrade the LAKE2.0 model and introduce a new variable, the water extinction coefficient for photosynthetically active radiation (PAR), to the model setup. During the initialisation, the model reads the available values of this coefficient and does a linear interpolation for every model timestep. Although the model results are not very sensitive to water extinction coefficient, the proposed modification allowed to improve the results in

some periods by about 1 degree as exemplified in the Fig. 2 for a selected period.

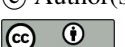



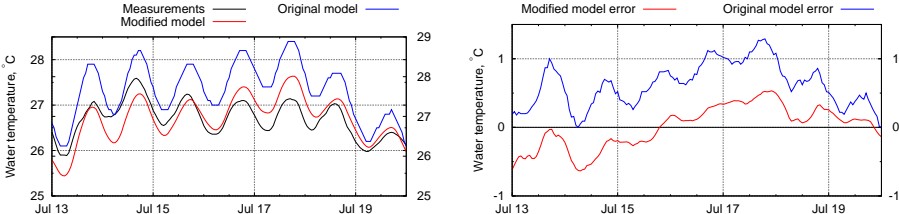

**Figure 2.** Water temperature in mixed layer, original and modified model results and their errors when compared to the observations made at the Montante platform.

## 2.5 FLake model

In addition to LAKE2.0, FLake model was used to simulate water temperature for chosen period. FLake model (Mironov, 2008) is based on a two-layer representation of the temperature profile and on the integral energy budgets for the two layers. The upper layer is assumed to be well mixed and the structure of the deep stratified layer is described using the concept of self-similarity of the temperature-depth curve. the mean water temperature during the simulation run and apart of it provides mixed layer and bottom temperature, thermocline shape factor, and mixed layer depth in the output. FLake model is widely used in climate and numerical weather prediction studies (Salgado and Le Moigne, 2010; Samuelsson et al., 2006; Le Moigne et al., 2016; Su et al., 2019) to simulate the feedback of freshwater lakes on the atmospheric boundary layer, and in the intercomparison experiments with other parameterizations. In particular, FLake has been applied in studies of the Alqueva reservoir by Iakunin et al. (2018), Potes et al. (2012), and Salgado and Le Moigne (2010).

## 2.6 Simulation setup

The simulation conducted in the present study covered 23 months from the 1st of May 2017 to the 29th of March 2019 with 1 hour timestep for input and output data. In the setup stage specific features of the Alqueva reservoir were prescribed: the series of PAR extinction coefficients for the simulation period, the morphometry of the lake bottom expressed via dependence of horizontal cross-section area on depth and the initial profiles of water temperature, $CO_2$, $O_2$, $CH_4$, and salinity (the last two profiles were set to zero due to the lack of the observation data).

Water pH significantly affects the solubility of carbon dioxide, but its value is a model scalar constant. In reality, observations show that pH tends to decrease near the bottom and has a seasonal variation, changing from 7.8 to 8.8 during the years (2017-2019) in the mixed layer. After averaging the measurements, pH constant inside the model code was altered from 6.0 to 8.48 for a better representation of real processes. Another modification has been done to the hypolimnetic diffusivity parameterization. According to Hondzo and Stefan (1993) for lakes of regional scale hypolimnetic eddy diffusivity rate $K_z$ is related to stability frequency $N^2$ and the lake area $A_s$:

$$K_z = c_1(A_s)^{c_2}(N^2)^{c_3},$$ (2)





where $c_1 = 8.17 \times 10^{-4}$, $c_2 = 0.56$, $c_3 = -0.43$ are empirical constants, $N^2 = -(\partial \rho / \partial z)(g/\rho)$, $z$ is depth, $g$ is acceleration of gravity, and $\rho$ is density of water. In LAKE2.0 model equation 2 is presented as $K_{z,LAKE2.0} = \alpha K_z$, where $\alpha$ is a calibration coefficient that allows to adapt this parameterization to the specific features of a given lake. In a series of sensitivity experiments it was found out that for simulation of thermal regime on the Alqueva reservoir the value of $\alpha = 0.3$ provides the best representation of the heat diffusion of heat from the surface to depth of the lake.

Both LAKE2.0 and FLake models were initialized with ALOP data measured at the Montante, in the reservoir floating platform and ran in standalone version. Atmospheric forcing input data were taken from the Montante platform observations. Gaps in data smaller than 3 hours were filled with a linear interpolation. In case of gaps bigger than 3 consecutive hours data was substituted with the corresponding values from the land weather stations (Barbosa and Cid Almeida). Comparison between LAKE2.0 and FLake models was made in terms of water temperature and heat fluxes over the water surface.

## 3 Results and discussion

### 3.1 Water temperature

Water temperature is a crucial factor for Numerical Weather Prediction (NWP) applications, for lakes vital activity, and their ecosystems. It is a key parameter of the lake-atmosphere interactions. Thus, detailed representation of the evolution of the water temperature at various depths is an important task.

Figure 3 shows the temporal evolution of the LAKE2.0 simulated water temperature in the reservoir over the whole chosen period. The summer period, characterized by a strong thermal stratification of water, is clearly seen on Fig. 3. It begins in late

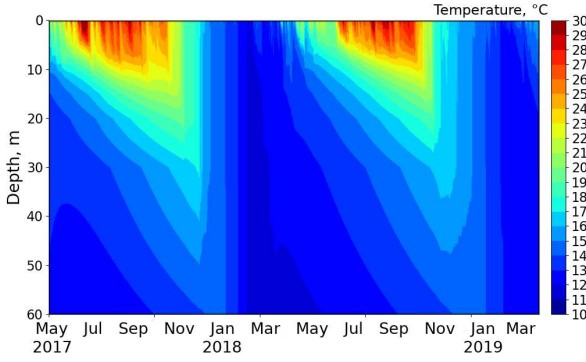

**Figure 3.** Time-depth Hovmöller diagram of the LAKE2.0 simulated water temperature in the Alqueva reservoir based on hourly data.

April and ends in November. Water temperature in these months in upper layers increases up to 30 °C and in the hottest months (July-September) reaches 25 °C at 10 m depth. Autumn turnover occurs in the end of October – beginning of November: water temperature becomes uniform at depths up to 30 meters and by the middle of December the lake shows no temperature stratification. During this period temperature decreases from 19 to 12 °C (in late February). In the end of April spring stratification occurs again and the cycle repeats.

The temperature of water in the mixed layer (ML) is of a particular interest in many studies. The LAKE2.0 provides water temperature at different depths defined in model setup, and ML thickness, assuming that ML temperature is constant (not including surface skin effect). As in the real mixing layer the temperature is not exactly constant, for the comparison we have chosen the water temperature at 50 cm deep. On the whole simulation period ML depth in the reservoir was never less than 70

cm. Figure 4 (a) shows LAKE2.0 simulated results in comparison with measured values and FLake results of ML temperature. To smooth hourly fluctuations in such long-term simulation, moving average was used with 6-hour period.

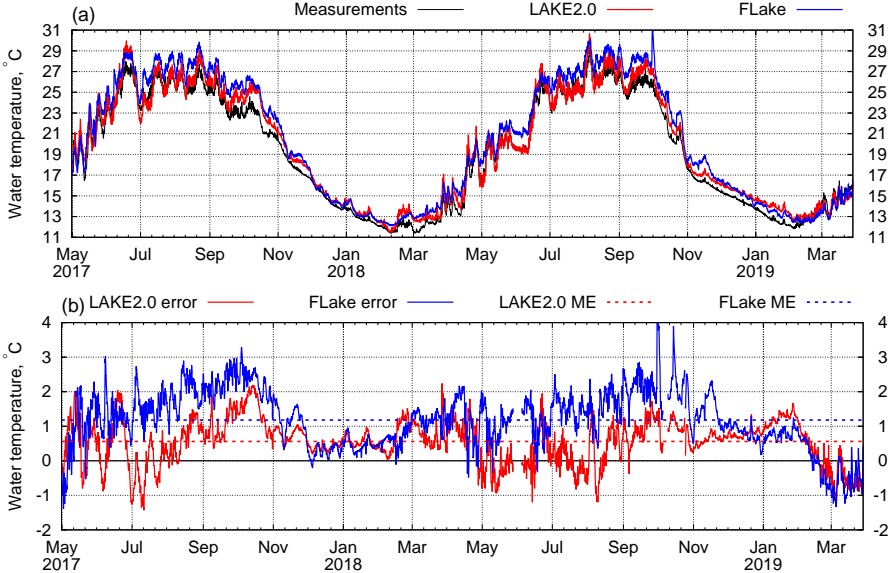

**Figure 4.** Top: time series of the Alqueva water temperature in mixed layer: measured (black curve), and modelled using the LAKE2.0 (red curve), and the FLake (blue curve). Bottom: errors of LAKE2.0 (red curve), FLake (blue curve) models relatively to the observations at the Alqueva reservoir.

Differences between the two model results and the measurements (errors) are shown in Fig. 4 (b). In the period of March-November of both years when the lake is stratified, the LAKE2.0 model demonstrates better results, while during the cold periods (November-March) both models shows similar error rates. Statistic of the comparison is presented in Table 2. Over-

all, mean absolute errors for the whole simulation period are 1.27 °C for FLake and 0.74 °C for LAKE2.0. Mean errors of the LAKE2.0 and FLake models for the simulation period are 0.56 and 1.18 °C correspondingly (shown as dashed lines in Fig. 4 (b)), which means that both models tend to slightly overestimate ML temperature. The LAKE2.0 model results are better for warm periods while FLake results are better for cold. Both models demonstrate almost identical correlation for the selected periods.

For more detailed analysis of the surface water temperature evolution we chose four months, July 2017/18 and January 2018/19, that represent stratified and non-stratified lake state to see the daily cycles of the ML water temperature (Fig. 5). It is seen that LAKE2.0 model shows exceptionally good results in summer months (Fig. 5 (a), average mean errors are -0.23 and





**Table 2.** Statistical results of ML water temperature intercomparison.

| Time periods | Correlation | | Mean error, °C | | MAE, °C | |
|---|---|---|---|---|---|---|
| | LAKE2.0 | FLake | LAKE2.0 | FLake | LAKE2.0 | FLake |
| May '17 – Oct '17 | 0.95 | 0.96 | 0.52 | 1.57 | 0.79 | 1.63 |
| Nov '17 – Feb '18 | 0.99 | 0.99 | 0.61 | 0.63 | 0.61 | 0.64 |
| Mar '18 – Oct '18 | 0.99 | 0.99 | 0.48 | 1.50 | 0.69 | 1.51 |
| Nov '18 – Feb '19 | 0.98 | 0.98 | 0.83 | 0.92 | 0.83 | 0.92 |
| All period | **0.99** | **0.99** | **0.56** | **1.18** | **0.74** | **1.27** |

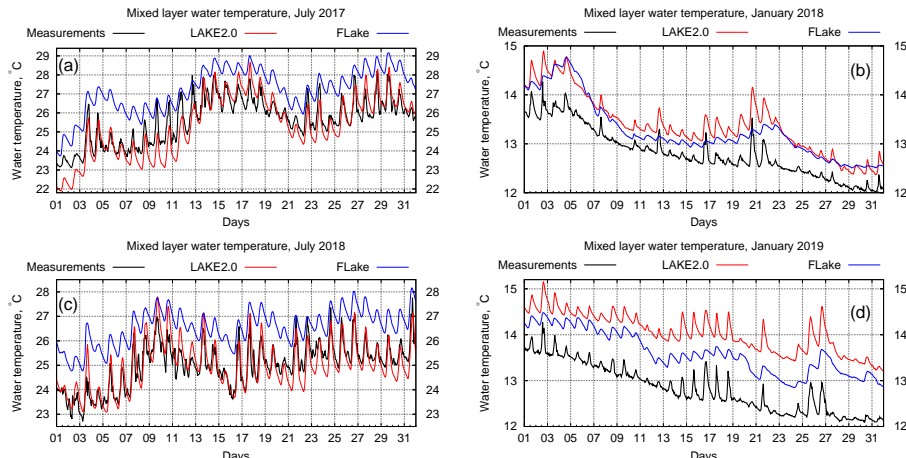

**Figure 5.** Timeseries of mixed layer water temperature for July 2017/2018 (a,c) and January 2018/2019 (b,d).

-0.04 °C for 2017 and 2018 correspondingly) while FLake provides an overestimation of 1-2 degrees and an underestimation of the daily amplitude. Correlation coefficients in this case are 0.94/0.88 (LAKE2.0) and 0.90/0.89 (FLake) correspondingly.

Diurnal ML temperature variations can reach 3 degrees and generally are well represented by LAKE2.0 model. In January the water temperature profile in the reservoir is homogeneous, daily amplitude is not so high (Fig. 5 (b)), and FLake model shows a smaller overestimation (0.95 correlation for both months and mean errors of 0.45/0.78 °C). LAKE2.0 results show a positive offset, average mean error for January 2018 is 0.78 °C and correlation is 0.97. In January 2019 LAKE2.0 mean error is 1.22 °C but, in general, the shape of the curve is similar to the measured and daily variations of temperature is represented quite good.

Temperature distribution with depth is another significant parameter for lake thermodynamics. LAKE2.0 model simulates water temperature at pre-defined depth levels; FLake provides a shape factor for the thermocline curve, ML and bottom temperature. Using these values it is possible to access the water temperature of thermocline beneath the ML at any depth. Simulation results are shown in Fig. 6 for the following cases: 15 July 2017, 15 January 2018, 15 July 2018, and 15 January 2019 on 12:00 UTC each.





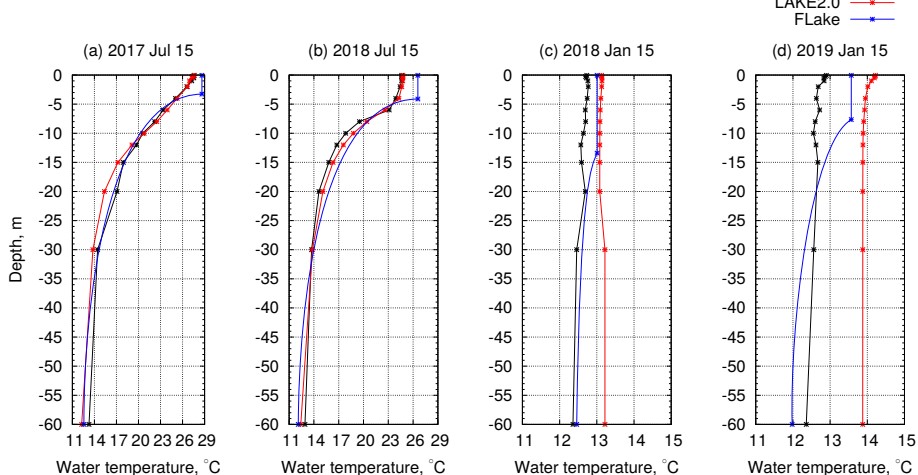

**Figure 6.** Water temperature profiles for 15 July 2017 (a), 15 January 2018 (b), 15 July 2018 (c), and 15 January 2019 (d), 12:00 UTC each.

Summer water temperature profiles are well represented by both models, although FLake shows an overestimation in the ML. In winter, on the other hand, LAKE2.0 overestimates water temperature through whole water column. Although LAKE2.0 reproduces the short-term (daily and weekly scales) thermal evolution of the ML very well, its integral energy is seemed to be higher than in reality. The errors are higher on the second year of the simulation in the results of winter 2018/19, exceeding 1 degree. The modelled water column tends to heat slightly more than the actual water column (Fig. 6 (c), (d)). This behaviour may be due to a small misrepresentation of the energy balance at the lake surface or at the bottom and requires additional tests that could eliminate such systematic errors and improve the results, especially in cold periods.

## 3.2 Heat fluxes

Sensible and latent heat fluxes play an important role in lake-atmosphere interaction, determining the rates of heat accumulation by water bodies or evaporation from the surface and consequently have effects on the local climate and on the establishment of thermal circulations (see for example Iakunin et al. (2018)). The LAKE2.0 model (as well as the FLake) is capable to calculate heat fluxes and the figure 7 represents the daily averaged results of the simulation of these variables.

Sensible heat flux is well represented by both models (Fig. 7 (a,b)) which is supported by low mean errors (see table 3) and high correlation coefficient. Latent heat flux, however, is overestimated by LAKE2.0 and FLake models (by 53-43 $\mathrm{Wm}^{-2}$) although both models demonstrate high correlation (0.92) with the measurements.

In terms of latent heat fluxes the LAKE2.0 model results are worse than the FLake when compared to the Eddy-covarience (EC) measurements. However, it should be noted that several studies have indicated that the eddy-covarience systems tend to underestimate the heat fluxes (e.g. Twine et al., 2000). Present results show comparable differences between the FLake and the LAKE2.0 models and EC measurements over lakes (Stepanenko et al., 2014; Heiskanen et al., 2015) in which the relative differences of about 35% were noticed. The differences between model and EC observations can also come from the model



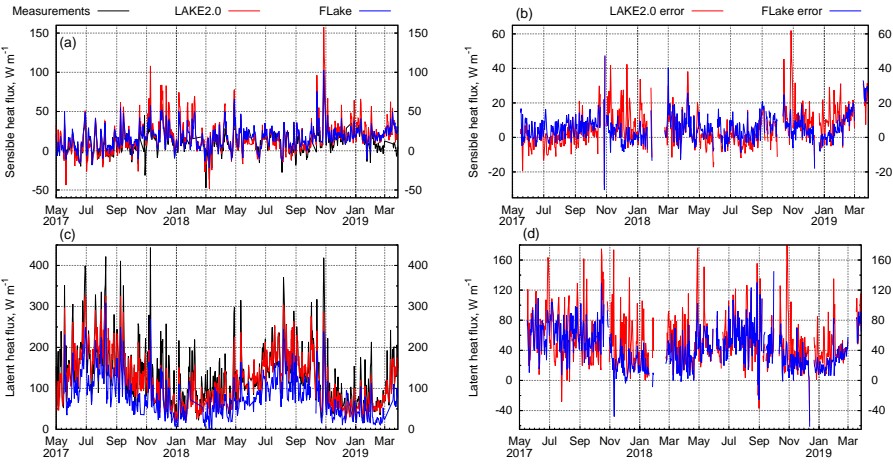

**Figure 7.** Daily averaged sensible (a) and latent (c) heat fluxes with corresponding errors (b, d). Black curve represents the measured values, red curve is associated with LAKE2.0 results, and blue curve is with FLAKE results.

**Table 3.** Sensible and latent heat flux errors and correlation coefficients

|  | Sensible heat | | Latent heat | |
| --- | --- | --- | --- | --- |
|  | LAKE2.0 | FLake | LAKE2.0 | FLake |
| Mean error, $Wm^{-2}$ | 5.51 | 5.36 | 52.93 | 43.46 |
| MAE, $Wm^{-2}$ | 8.38 | 6.85 | 53.40 | 44.02 |
| Corr. coefficient | 0.88 | 0.87 | 0.92 | 0.92 |

errors due to the fact that the Alqueva reservoir is an open lake with a constant inflow and outflow of Guadiana river. The horizontal flows, not represented in one-dimensional vertical models, can add or remove energy from the water body. Also, the water level of the Alqueva changes significantly during the year due to drought periods and discharges through the dam. It decreased on up to 7 meters in 2018 that corresponds the loss of 35% of total volume of water. The models cannot take into an account those changes while they could be a major source of errors in heat flux computations.

**3.3   Dissolved carbon dioxide**

The diffusion of $CO_2$ from the atmosphere to water and its further dissociation are of major importance to photosynthetic organisms which depends on the availability of inorganic carbon (Wetzel, 1983). Dissolved inorganic carbon constituents also influence water quality properties such as acidity, hardness, and related characteristics.

    The solubility of $CO_2$ in water depends on several factors such as pH, water temperature, etc. Observations indicate that

pH may vary from 8.8 at the surface level to 7.4 at the bottom, while in the model it is a constant parameter which value



was set to 8.48 which corresponds to the mean pH value during the simulation period. Figure 8 reveals the dynamics of $CO_2$ concentration on water in the first months of the ALOP field campaign in comparison with LAKE2.0 simulated results.

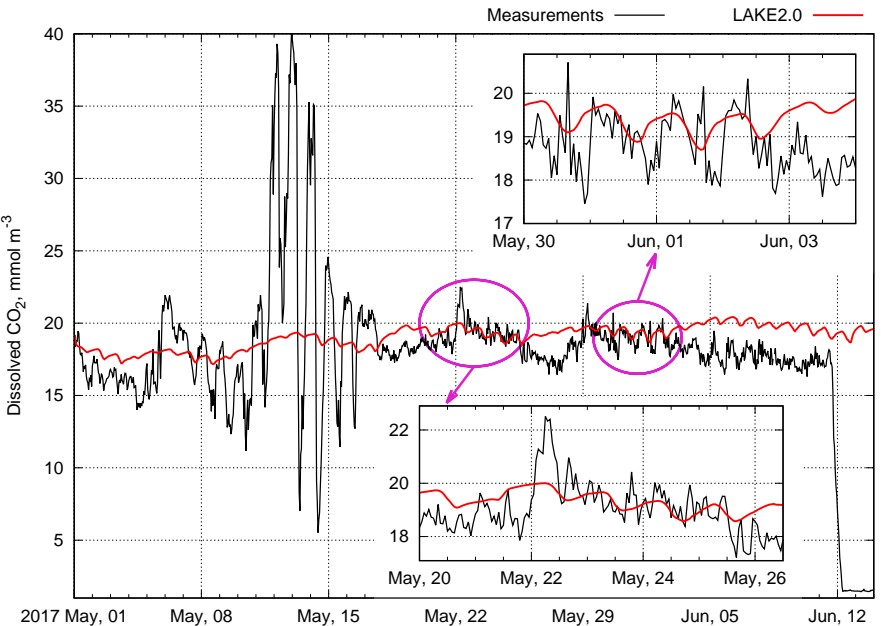

**Figure 8.** Timeseries of dissolved $CO_2$ in water at 25 cm depth

In general, the LAKE2.0 values are smoother than the observations as the model does not react to the changes in $CO_2$ so fast but the mean values are well represented. On May 20-26 and in the beginning of June (subplots in Fig. 8) daily cycles are represented quite good. In the second week of May, $CO_2$ probe accidentally dismounted from the platform and remained floating in the water on the connecting cord until the next field work trip (17th of May). This explains the rapid changes in measured values in this period. On the 12th of June the probe failed and it was dismounted and removed from the Montante platform. Later on the 18th of October the probe was mounted on the platform again and it was working in a test mode for three weeks (Fig. 9). In this period LAKE2.0 simulated values of $CO_2$ do not show much daily variations and have an increasing trend due to autumn water cooling. Small daily biases in simulated values coincide with peaks in measured data.

Thus, we can conclude that in long-time simulations LAKE2.0 model represents $CO_2$ trends quite well. The model failed to reproduce diurnal cycle of the surface carbon dioxide concentration which calls for inquiry of parameterizations of photosynthesis and respiration in the model. However, the diurnal means are well captured which is enough in perspective of using the model in climate applications.

## 3.4 Dissolved oxygen

Dissolved oxygen (DO) is essential to all aerobic organisms living in lakes or reservoirs. To understand the distribution, behaviour, and growth of these organisms it is necessary to know the solubility and dynamics of oxygen distribution in water.

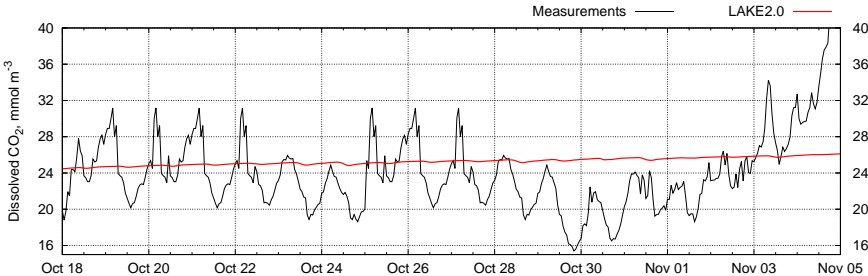

**Figure 9.** Timeseries of dissolved $CO_2$ in water at 25 cm depth for the period 18 October – 5 November.

The rates of supply of DO from the atmosphere and from photosynthetic inputs, and hydromechanical distribution of oxygen are counterbalanced by consumptive metabolism. The rate of oxygen utilization in relation to synthesis permits an approximate

evaluation of the metabolism of the lake as a whole (Wetzel, 1983).

The concentration of DO in the Alqueva reservoir was measured continuously on the Montante platform since July 3rd 2018. Comparison of measured and model values are shown in Fig. 10. The model represents DO concentration in a realistic way

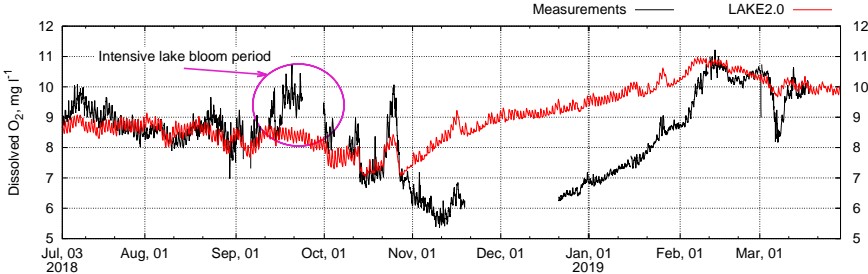

**Figure 10.** Timeseries of dissolved $O_2$ in water at 25 cm depth for the period July 3, 2018 — March 29, 2019 at the Alqueva reservoir.

during the first two months until the middle of September when a microalgal bloom occurred. It caused an intensive production of $O_2$ in water that can not be represented by the LAKE2.0 which does not have an explicit representation of algae, and the

bloom does not affect atmospheric forcing. Then, until the end of October, the model showed good results but in November the observations demonstrated a decrease of oxygen concentration which was not followed by the model, in fact, the model predicted an increase until the beginning of February. In November the turnover happens, water temperature decreases and does not change significantly with depths; under this conditions concentration of oxygen producing organisms decreases so does DO that falls from 8-9 to 6 mg $l^{-1}$. The model does not reflect this decrease in photosynthesis but largely increases DO

concentration following the decrease of water temperature (oxygen is more soluble in colder water). When in the middle of February temperature returns to stratified regime DO concentration in the model and measurements coincide again.

The photosynthesis rate can be linked to chlorophyll-a measurements (Table 4) which were done during field work at the Alqueva reservoir. In July 2018, when DO measurements began, concentration of chlorophyll-a ranged from 1.754 to





2.98 mgl$^{-1}$ in water ML (0-3 m). Further, when autumn bloom occurred in September, chlorophyll concentration significantly

increased and reached 14.036 mgl$^{-1}$ at the surface, and came back to values of 2.309 mgl$^{-1}$ in November. LAKE2.0 model solves DO concentration assuming a constant chlorophyll-a concentration of 2.3 mgl$^{-1}$. ALOP field campaign ended in December 2018 but the work on stations and the Montante platform maintenance continued, so in January and February 2019 samples from water surface layer were taken. The sample of January 15 showed no traces of chlorophyll-a in water which is related to very low DO concentration in this period (Fig. 10). The measurements of chlorophyll-a in water sample taken on 2nd

of February showed the value of 1.3 mgl$^{-1}$. It corresponds to the relative increase of oxygen producers in water, and hence, DO concentration.

Analysis of DO profiles (Fig 11) shows similar results. Distribution of oxygen with depth are well represented by the model for July and September profiles, while in December and February with no stratification in temperature and oxygen LAKE2.0 model overestimates DO on up to 2.5 mg l$^{-1}$. March profiles (1 and 29) show good similarity in measured and simulated

values.

## 4 Conclusions

Numerical studies of the seasonal variations of the thermal and gas regimes in the Alqueva reservoir with using the LAKE2.0 and the FLake models are presented in this work. Simulated profiles and timeseries of water temperature, sensible and latent heat fluxes, concentrations of dissolved $CO_2$ and $O_2$ were compared with observed data. The seasonal variations of the ML

water temperature are well represented by both models. Mean absolute errors are 0.74 °C and 1.27 °C for LAKE2.0 and FLake models correspondingly and the correlation coefficients are 0.99 for both. The LAKE2.0 model overestimates ML water temperature only by 0.5 °C during the warm periods (March – October), while FLake shows overestimation about 1.5 degrees. In the cold periods (November – February) both models show the same rate of overestimation of ML temperature about 0.6-0.9 °C.

The model errors of the seasonal variations in sensible and latent heat fluxes are the following. Sensible heat MAEs are 7.71 Wm$^{-2}$ (LAKE2.0) and 6.75 Wm$^{-2}$ (FLake). Latent heat flux results of both models in terms of MAE are worse: 53.99 Wm$^{-2}$ (LAKE2.0) and 45.6 Wm$^{-2}$ (FLake). Such errors could be due to the sporadic input of measured wind speed, which values change rapidly.

**Table 4.** Chlorophyll-a measurements at the Alqueva

| Depth | Chlorophyll-a concentration, mg l$^{-1}$ | | | | | | | | | | |
|---|---|---|---|---|---|---|---|---|---|---|---|
| | '17 Jul | Sep | Nov | '18 Jan | Apr | Jun | Jul | Sep | Nov | 19' Jan | Feb |
| Surface | 1.11 | 7.60 | 1.03 | 2.55 | 12.189 | 5.796 | 2.678 | 14.036 | 2.309 | No pigments | 1.2 |
| 1 m | 0.00 | 6.33 | 0.78 | 2.33 | 12.695 | 4.344 | 1.754 | 6.279 | 1.385 | — | — |
| 2 m | 1.11 | 6.65 | 1.03 | 2.44 | 11.573 | 3.989 | 2.124 | 7.849 | 1.847 | — | — |
| 3 m | 2.77 | 6.65 | 0.96 | 1.99 | 9.973 | 3.022 | 2.980 | 9.603 | 1.385 | — | — |





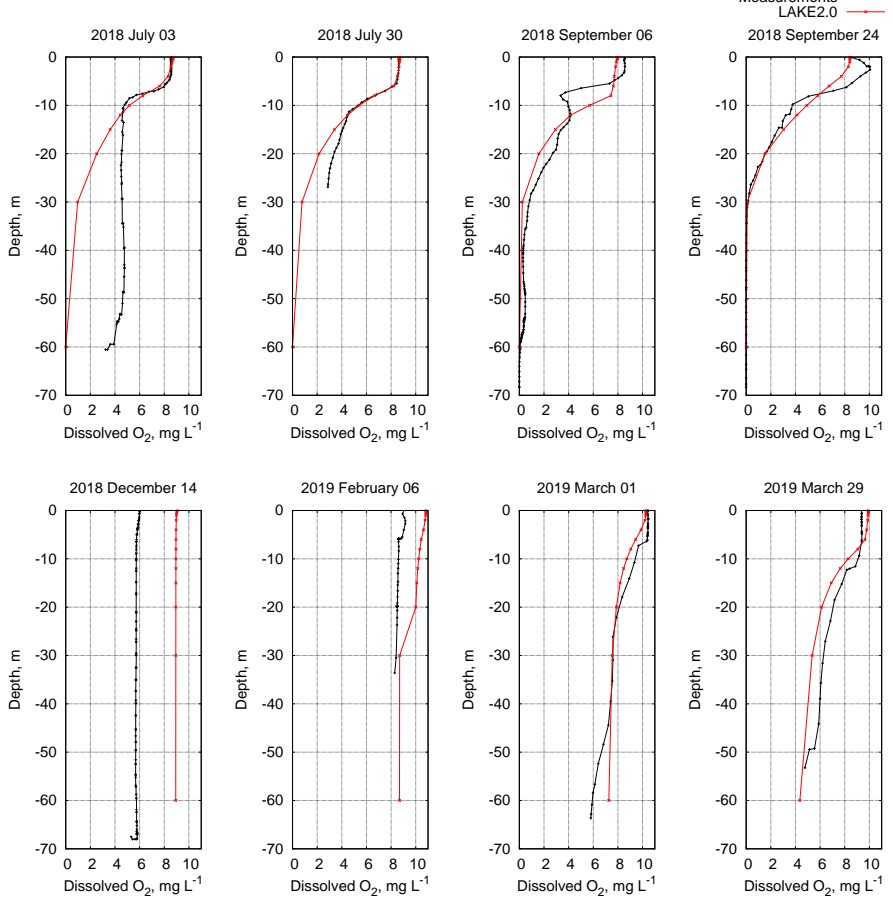

**Figure 11.** Profiles of dissolved $O_2$ in water measured during the field campaign (Black) and model values (red).

LAKE2.0 simulated dissolved carbon $CO_2$ timeseries demonstrated a good accordance with the observations in mean values,
however the model significantly underestimated the magnitude of diurnal cycle. On the second year of the experiment (October 2018, when the probe was returned to the platform), simulated $CO_2$ values did not show big errors despite the fact that pH value remained constant during the whole simulation period.

Dissolved oxygen, reproduced by the model, reveals the need of modernisation of LAKE2.0 model before operational use. Although the simulated values of $O_2$ may concur with measurements in short time intervals, annual Alqueva oxygen cycle
cannot be reproduced because the model does not respond to algae bloom (underestimation of $O_2$ values) and winter minimum (high overestimation). Winter overestimation is supposedly due to the relatively low water temperatures. Nevertheless, high versatility and flexibility of the LAKE2.0 model gives good opportunities for elimination this flaws with the aim of adequate modelling of seasonal variations in gas regime of the lake.

The performed simulation confirms that the FLake model is a good option to be used to forecast lake surface water tempera-
ture namely in Numerical Weather Prediction in which the running time is critical. The results are encouraging as to the ability





of the LAKE2.0 model to represent the evolution of physicochemical profiles inside the lakes, and may be used operationally in the future, coupled with weather prediction models, to forecast variables useful in the management of water quality and aquatic ecosystems. Similarly, the results indicate that the LAKE2.0 model could be used in climate modelling to estimate the impacts of the climate change on the thermal and gas regimes of the lake.

*Code and data availability.* The current versions of the models used in the work as well as the atmospheric forcing data can be obtained at *https://doi.org/10.5281/zenodo.3608230* or upon request from the author (Maksim Iakunin, *miakunin@uevora.pt*, *m.yakunin89@gmail.com*). The source code of the FLake model is available for download at the website (*http://www.flake.igb-berlin.de/site/download*). The source code of the latest versions of the LAKE2.0 model is available at the website (*http://tesla.parallel.ru/Viktor/LAKE/wikis/LAKE-model*).

*Competing interests.* The authors declare that they have no conflict of interest.

*Acknowledgements.* The work is co-funded by the European Union through the European Regional Development Fund, included in the COMPETE 2020 (Operational Program Competitiveness and Internationalization) through the ICT project (UID / GEO / 04683/2019) with the reference POCI-01-0145-FEDER-007690 and also through the ALOP project (ALT20-03-0145-FEDER-000004). Victor Stepanenko was supported by Russian Science Foundation (grant 17-17-01210).



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
