# Peer review of "Numerical study of the seasonal thermal and gas regimes of the large artificial reservoir in Western Europe using LAKE2.0 model"

_Geoscientific Model Development, 2019_

## Referee Comment (RC1) · Anonymous Referee #1 · 5 Mar 2020

Review for Journal: GMD Title: Numerical study of the seasonal thermal and gas regimes of the large artificial reservoir in Western Europe using LAKE2.0 model Author(s): Maksim Iakunin et al. MS No.: gmd-2019-309

In theri paper the authors show measurments and numerical simulation results with the two layer model FLake and the onedimensional model LAKE2.0. The authors compare temperature and heat fluxes and dissolved gases such as $CO_2$ and $O_2$. In general, the results of Lake".0 are better than FLake. Thsi is no surprise and FLake is a simpler approach. Lake2.0 reproduces temperatures in general with an error of 1°C. The simulation of dissolved gases in general is good. However the daily variation is missing

because the model does not reflect biological activity completely enough and during deep recirculation modelled oxygen lies higher than measured.

The accuracy of temperature stratification of 1°C indicates careful set-up of the model, and can only be achieved by considering the variable light absorption during the year. The good agreement between modelled dissolved oxygen and measurements during the stratification period indicates that the oxygen concentration in the surface layer is controlled mainly by exchange with the atmosphere, as biological processes are not reflected in the model; as a consequence during winter and during strong blooms the modelled oxygen is not as good anymore. I suspect there could be an upwelled oxygen demand during deep recirculation that is not included in the model.

This manuscript is significant as it shows that temperature of reservoirs can well be represented with 1_d models: Lake2.0 is an option. Modelling oxygen and carbon dioxide is possible with simple assumptions (without biological model); however for representation of oxygen during deep recirculation and during algal blooms as well the daily variation of CO2, a more detailed model is necessary: Scientific significance 2.3; Scientific quality: the modelling is well done; the improvement of absoption was a smart step; the manuscript gives a competent impression; measurments suited for checking simulation: scientific quality 1.6. Scietific reproducibility. 1,5 see below Presentation quality: 2; English writing: in general good: see comments below. the fonts in Figures are generally too small: especially: Figs. 1, 2, 5, 7; number of figures is appropriate.

Important issues: 1) I would recommend not to use a contour plot to judge onset and end of stratification: the authors claim startfication starts iend of April: Fig.11 indicates clearly, the reservoir was stratified on 2019 March 1st and 29th. An earlier onset of stratification explains the better oxygen simulation in Fig 10 from ca. Feb 8th. Better look at temperature tracks of sensors of different depths. 2) At two places the pH dependence of CO2 solubility is mentioned, e.g. line 177. Do the authors indicate the Henry coefficient depends on pH; or does this refer tot he much stronger pH dependence oft he carbonate equilibrium? Hs bicarbonate been modelled, or DIC

(doissolved inorganic carbon)? – connected with this: I cannot realy follow what the altered pH on line 179 may indicate. 3) There could be a mention oft he two most commen 1D lake models: DYRESM (Imberger) and GLM (e.g. Bruce et al. of Hipsey et al) Technical details: Line 8: capable OF . . . Line 98 : BEING used . . . Line 125: are -> is Line 137 "into account" to the end of phrase Line 147. Check sentence Line 155: unit is missing Line 162: THE chose period Line 165 capital T Line 173: comma after stage? Line 185: how is density rho calculated (including solutes?) Line 191: comma after hours (possibly also before bigger) Caption Fig. 4: include the information of 6-hours mean Caption Fig. 4: errors-> temperature difference Line 221: that -> which Line 237. Remove "is" Line 251: covariAnce Line 255: constant -> continuous Line 266: which WAS SET TO A VALUE OF 8.48, which correspondED . . . Figure 8: remove useless data between 10 and 17 March (or explain what can be seen) Line 274 comma after period Line 270: good -> well
* * *

---

## Referee Comment (RC2) · Anonymous Referee #2 · 30 Mar 2020

General comments

It is useful that this paper presents a model comparison that focuses on the factors that would be most important in influencing the heat and gas fluxes from a lake. It was good to see model comparisons focusing on the mixed layer. However, there is a need to clearly state the criteria used to define the mixed layer. I was pleased to see the specific comparison of measured heat fluxes and gas fluxes collected at high resolution with simulated data from two models. This not commonly done and is a unique and valuable aspect of this paper. And for these reasons I think this paper does document important progress in lake model development and does deserve to be accepted for publication

following revision.

I think for a modeling study such as this there is a need for more information on calibration. How the model was calibrated and what the final error levels were. It doesn't need to be extensive but as a minimum I would like to see a listing of the final calibrated parameter values, as well as a brief description of what each parameter does. A scatter plot of the simulated vs measured temperature. And some statistics on model fit (ie RMSE , MAE etc ) Furthermore, I'm assuming that the model was calibrated against measurements of water temperature, but this may not entirely be the case since there were measurements of gas concentrations and heat fluxes that could in theory also be used for calibration. What was used for calibration should be clearly stated.

One of the most important aspects of this paper is the comparison between the simulated gas fluxes with measured data. Therefore, I do think there is a need to better describe the equations governing $CO_2$ and $O_2$ concentrations. I was not that familiar with Lake 2.0 but after searching a bit I found that this is not the first time $CO_2$ and $O_2$ have been simulated with Lake 2.0, even though this paper may be one of the best verification studies. I would like to see some overview description of the main processes affecting the $CO_2$ and $O_2$ concentrations and also more clear references to the original publications where the equations describing these processes are completely defined

There were two things that were changed in the version of the model used in this study (the fixed pH value and the equations affecting diffusion in the hypolimnion) and also one assumption (fixed chlorophyll concentration) which I suspect and which later in the paper the authors also suspect leads to errors in the simulated oxygen concentration. I think all three of these should be evaluated in a sensitivity analysis as part of the paper in section 2.4 as was done with the light extinction coefficient.

There are quite a few small language errors in the paper. I have tried to suggest solutions to many in the technical comments. These should not be allowed to take away from the good scientific and technical aspects of this study, so I think it would be

good to have paper carefully proofread for language before the final submission

Specific comments

Abstract - Since you mention the Flake model in the abstract I think you should have a brief statement about how well it worked compared to Lake2.0

Line 40 - what do you mean by "to complete the results". Does Flake do something that Lake2.0 does not? Or are you comparing the results of the two models

Lines 98-99 I think you could give a bit more information. What type of errors? How much missing information was there? Linear interpolation?

Line 104 Are not these fluxes also occurring through the surface?

Line 105 and unlike Hostetler model I don't know what you mean by unlike Hostetler model. Are you using this model as well? Or our components of this model embedded in Lake2.0?

Line 141 the description of photosynthesis is rather unusual. Is it really reasonable to assume that chlorophyll remains constant while photosynthesis is changing? Perhaps this simplification can be justified by the fact that the modeling is mainly looking at gas exchange and not the biology of the lake. However, photosynthesis will affect both O2 and CO2. Assuming a constant chlorophyll concentration could greatly under or over estimate the total photosynthesis in the epilimnion. I think there should be more justification for the constant chlorophyll assumption. Perhaps a sensitivity analysis on how changes in chlorophyll affect the gas flux estimates.

Later in lines 150-160 you document a large seasonal variation in the attenuation co-efficient. How much of this is due to changes in chlorophyll? Could this variability invalidate the assumption of a constant chlorophyll concentration? Also in this section the model was modified to allow the input of a varying extinction coefficient which is a good idea. However, this could be described more clearly. It is stated that "introduce a new variable,the water extinction coefiňĄcient for photosynthetically active radiation

(PAR), to the model setup" How does this coefficient differ from the coefficients described in line 120. Perhaps you mean that the existing coefficient described in line 120 was changed from a fixed model coefficient to a time varying one? However Im still a little confused since PAR is usually considered to be between 400-700 nm and measured as a photon flux density, whereas I would think that the coefficient described on line 120 would have a wider bandwidth and would be measured in terms of watts

Lines 165-166 need to be made clearer.

Starting at line 177 there are two changes to the model described one concerning pH and the other concerning hypolimnetic diffusivity. Sensitivity analysis should be done for both of these and these results would be better presented in the section starting on line 150

Line 191 this information is better place in the section on observational data (see comment above).

Line 207 Model simulations of the mixed layer depth (MLD) are discussed. The method for defining the MLD should be described in the methods section. Also in figure 4 it would be good to show a plot of the variations in the MLD over time. In the caption of fig 4 describe what the dashed horizontal lines represent.

Line 232 states "water temperature of thermocline beneath the ML at any depth" Don't you mean water temperature of the hypolimnion?

Line 252 states "Present results show comparable differences between the FLake and the LAKE2.0 models and EC measurements over lakes (Stepanenko et al., 2014; Heiskanen et al., 2015)" To me present studies means the study being described in this paper. Do you mean something like other recent studies?

Lines 255-259 Is it possible to estimate the depth of the horizontal flows from the surface temperature of the inflowing river? It seems like this would be most significant if the inflows are moving through the surface layer.

Line 270 In the second week of May, CO2 probe accidentally dismounted from the platform and remained. - I think you should just remove these data from the plot You clearly do not believe they are meaningful and have a good explanation for this.

Lines 298-299 Chlorophyll concentrations are given in mg/l should these be ug/l (10-6g/l)? The mg/l concentrations that are given are very high and would be considered typical of a highly eutrophic lake. They would also certainly greatly affect the O2 concentration I have the same concern for the values in table 4.

Line 322 You state "Such errors could be due to the sporadic input of measured wind speed, which values change rapidly" First I would suggest sporadic nature rather than input. But I also think this needs more of an explanation Are we talking about errors in both latent and sensible heat? And what is the mechanism by which sporadic winds are increasing model error?

Line 325 You state "On the second year of the experiment (October 2018, when the probe was returned to the platform), simulated CO2 values did not show big errors despite the fact that pH value remained constant during the whole simulation period." But was not the pH also constant during the first year? Were there larger errors in the first year due to the fixed pH?

Line 334 This final paragraph needs to be reworked First I think you should be stressing that the Lake 2.0 model was shown to accurately simulate the heat fluxs and gas fluxes from the ML. I think this is one of the major model developments being described here. Secondly, I don't think you should start out by say that Flake is good model – Im sure this is true but it is not the purpose of this paper. You should be stating that Lake 2.0 is as good or better than Flake as you have shown in some of the comparisons in the paper. Finally in terms of using these two models to improve weather predictions you state that Flake has lower computational demands. By why not give some numbers on this? How much slower is Lake 2.0? Is it realistic to think it could be used to support weather prediction in the future?

Technical corrections

Line 1 (Suggested change) The Alqueva reservoir (southeast of Portugal) being the largest artificial lake in Western Europe and a strategic freshwater supply in the region. The reservoir is of scientific interest and monitored in order to maintaining the quality and quantity of water and evaluate its impact on the regional climate. To support these tasks we conducted numerical studies of the thermal and gas regimes in the lake

(Suggested change)supplemented by the data observed at the weather stations and the floating platforms deployed during the field campaign of the ALOP (ALentejo Observation and Prediction System) project. One-dimensional model LAKE2.0 was used for the numerical studies

line 8 this parameterization » this model?

Line 14 particpants » regulators

Line 20 allow to use them » allows them to be used

Line 25 models are important models is important

Line 30 for the lake ecosystem vital activity » regulating lake ecosystem processes

Line 39 allowing to reproduce the concentrations » that simulates the concentrations

Line 43 spelling (hypolimnion) forced with the observed data » forced with the observed meteorological data

Line 49 spreading along 83 km over » spreading over 83 km of of Guadiana » of the Guadiana

Line 51 the capacity of water » the storage capacity of water

Line 59 in favourable position. » into a favourable position. Rainfall seasons normally last from » Seasonal rainfall normally occurs between

[Figure]

Lines 64-65 Geographical and climatological factors make the Alqueva reservoir a vital source of fresh water needed to support the population and economy in the region, while on the other hand, an increasing anthropogenic

Line 71 air columns, over the water-atmosphere interface, and in the shores » air columns, at the water-atmosphere interface, and on the shores

Line 73 4 floating platforms » four floating platforms

Line 76 was settled on the platform » was deployed on the platform

Line 77 an eddy-covariance system, Campbell Scientiﬡc Irgason, provides data for atmospheric » an eddy-covariance system, Campbell Scientiﬡc, provides data of atmospheric

Line 89 and for punctually vertical proﬡles » and was occasionally used to collect vertical profiles

Line 94 were obtained in automatic regime and transferred » were automatically downloaded and transferred

Line 96 weather stations and conduct measurements, to collect » weather stations, to conduct more detailed measurements, and to collect

Line 125 condition »conditions

Line 126 condition is»conditions are

Line 161 (Suggested change) In addition to LAKE2.0, The FLake model was used to simulate water temperature for the chosen period. FLake model (Mironov,

Line 162 a two-layer representation of the temperature proﬡle » a two-layer representation of the lake's thermal structure

Line 196 (Suggested change) Water temperature is a crucial factor for Numerical Weather Prediction (NWP) applications, and as a regulator of lake ecosystem activity, and their ecosystems.

Line 237 its integral energy » the simulated heat content of the entire water column.

Line 245 is capable to calculate » are capable of calculating

Line 246 and the figure 7 represents » and figure 7 shows.

Line 270 are represented quite good » are represented quite well

Line 292 In November the turnover happens » In November, following turnover

Line 312 Alqueva reservoir with using the LAKE2.0 » Alqueva reservoir using the LAKE2.0

Line 316 correlation coefficients are 0.99 for both » correlation coefficients for the relationship between simulated and measured temperature are 0.99 for both

Line 317 FLake shows overestimation about 1.5 » FLake shows an overestimation of about 1.5

Line 318 show the same rate of overestimation » show the same level of overestimation

Line 324 good accordance » good corrospondance.

Line 328 of modernisation of LAKE2.0 » inclusion of a more complete description of the process regulating photosynthesis and respirations in the LAKE2.0 model

Line 329 (Suggested change) Although measured oxygen concentrations are well simulated values of O2 over short time intervals, the annual Alqueva oxygen cycle cannot be reproduced because the model does not respond to changes in algal concentration (underestimation of O2 values) and winter minimum (high overestimation). Winter overestimation is supposedly due to the relatively low water temperatures.

Why above is it supposedly? Are you not sure?

[Figure]

2020.

---

## Author Comment (AC1) · 11 May 2020

**Numerical study of the seasonal thermal and gas regimes of the large artificial reservoir in Western Europe using LAKE2.0 model**

AUTHORS' RESPONSES TO THE REFEREE #1 COMMENTS

Maksim Iakunin[1], Victor Stepanenko[2], Rui Salgado[1], Miguel Potes[1], Alexandra Penha[3,4], Maria Helena Novais[3,4], and Gonçalo Rodrigues[1]

*miakunin@uevora.pt*

[1]Department of Physics, ICT, Institute of Earth Sciences, University of Évora, 7000 Évora, Portugal
[2]Lomonosov Moscow State University, GSP-1, 119234, Leninskie Gory, 1, bld. 4, Moscow, Russia
[3]Water Laboratory, University of Évora, P.I.T.E. Rua da Barba Rala No1, 7005-345 Évora, Portugal
[4]Institute of Earth Sciences — ICT, University of Évora, Rua Romão Ramalho 59, 7000-671 Évora, Portugal

**Contents**

**Introduction. Document structure**

This document contains authors' responses to the comments of the Anonymous Referee. The document structure is the following:

- Referee's comments are numbered and given in *italic font*. General, specific, and technical comments come separately.

- Authors' response follows the comment and starts after **"Response:"** with normal font.

- The text from the article itself (if some changes are done, and if it is reasonable to provide it) is typed with `typewriter font` and separated from the response with an extra blank line.

- *Technical comments and mistakes* are not numbered, and authors' response follows immediately.

Reviewed manuscript with all the corrections is given after all responses. It contains the changes and proposals of **two** Referees and was prepared using LaTeXdiff package for better understanding of what has been changed.

**Anonymous Referee #1**

**General comments**

*In theri paper the authors show measurments and numerical simulation results with the two layer model FLake and the onedimensional model LAKE2.0. The authors compare temperature and heat fluxes and dissolved gases such as CO2 and O2. In general, the results of Lake2.0 are better than FLake. Thsi is no surprise and FLake is a simpler approach. Lake2.0 reproduces temperatures in general with an error of 1°C. The simulation of dissolved gases in general is good. However the daily variation is missing because the model does not reflect biological activity completely enough and during deep recirculation modelled oxygen lies higher than measured.*

*The accuracy of temperature stratification of 1°C indicates careful set-up of the model, and can only be achieved by considering the variable light absorption during the year. The good agreement between modelled dissolved oxygen and measurements during the stratification period indicates that the oxygen concentration in the surface layer is controlled mainly by exchange with the atmosphere, as biological processes are not reflected in the model; as a consequence during winter and during strong blooms the modelled oxygen is not as good anymore. I suspect there could be an upwelled oxygen demand during deep recirculation that is not included in the model.*

*This manuscript is significant as it shows that temperature of reservoirs can well be represented with 1 d models: Lake2.0 is an option. Modelling oxygen and carbon dioxide is possible with simple assumptions (without biological model); however for representation of oxygen during deep recirculation and during algal blooms as well the daily variation of CO2, a more detailed model is necessary: Scientific significance 2.3; Scientific quality: the modelling is well done; the improvement of absoption was a smart step; the manuscript gives a competent impression; measurments suited for checking simulation: scientific quality 1.6. Scietific reproducibility. 1,5 see below Presentation quality: 2; English writing: in general good: see comments below. the fonts in Figures are generally too small: especially: Figs. 1, 2, 5, 7; number of figures is appropriate.*

**Response:**    We thank the Reviewer for the positive comments about the article. The paper was re-edited very carefully and modifications and improvements were made. Below, we address every comment and explain the corresponding changes in the manuscript.

**₅ Important issues**

**Comment 1**

*I would recommend not to use a contour plot to judge onset and end of stratification: the authors claim startfication starts iend of April: Fig.11 indicates clearly, the reservoir was stratified on 2019 March 1st and 29th. An earlier onset of stratification explains the better oxygen simulation* ₁₀ *in Fig 10 from ca. Feb 8th. Better look at temperature tracks of sensors of different depths.*

**Response:**    We carefully analysed the water temperature data and used the definition of lake summer stratification given in Wetzel R. G. *Limnology*, Saunders College Publishing, 2nd edition, 1983, p 75, which implies it when a stratum of thermal discontinuity exists between epilimnion and hypolimnion (usually accepted as a change of $>1°C$ per metre. These led us to rework the ₁₅ second paragraph of section **3.1** concerning the stratification periods and update Fig. 3. However, we consider the figure with time-depth temperature diagram more demonstrative than time series of water temperature on various depth. The reworked part is the following:

According to the definition given in (Wetzel, 1983), summer stratification period is char- ₂₀ acterized by a stratum of thermal discontinuity (metalimnion) which separates an upper layer of warm circulating water (epilimnion) and cold and relatively undisturbed water below (hypolimnion). The stratum of thermal discontinuity is usually defined as a change of $>1°$ C per metre. Summer stratification periods are clearly seen in Fig. 1 (marked with dashed lines). The simulation began in a stratified conditions which lasted until 3 October 2017 while in ₂₅ 2018 stratification lasted from 14 April to 19 September.

[Figure]

Figure 1: Time-depth Hovmöller diagram of the LAKE2.0 simulated water temperature in the Alqueva reservoir based on hourly data. Dashed lines indicate the end (black) and the beginning (red) of stratification.

**Comment 2**

*At two places the pH dependence of CO2 solubility is mentioned, e.g. line 177. Do the authors indicate the Henry coefficient depends on pH; or does this refer tot he much stronger pH dependence oft he carbonate equilibrium? Hs bicarbonate been modelled, or DIC (doissolved inorganic carbon)?*
5   *– connected with this: I cannot realy follow what the altered pH on line 179 may indicate.*

**Response:**   We decided to add a supplementary material to the article where details on the biogeochemical processes are provided widely. You may find it attached to this file after the article.

Evolution and vertical distribution of three dissolved gases are considered in the LAKE2.0
10   model, which are methane $CH_4$, oxygen $O_2$ and carbon dioxide $CO_2$. However, dissolved carbon dioxide is supposed to be always in carbonate equilibrium, so that it contributes to concentration of dissolved inorganic carbon (DIC), $C_{DIC} = C_{CO_2} + C_{HCO_3^-} + C_{CO_3^{2-}}$, and it is the change of DIC that reflects the number of carbon atoms in $CO_2$ molecules added to (or lost by) a solution from (to) atmosphere, bubbles, respiring organisms or decaying organical
15   matter (see Section 1.2 in Supplementary).
    In addition, the content of dissolved organic carbon (DOC), particulate organic carbon (both living, POCL, and dead, POCD) are calculated. POCL includes carbon atoms contained in phytoplankton and zooplankton.

**Comment 3**

20   *There could be a mention oft he two most commen 1D lake models: DYRESM (Imberger) and GLM (e.g. Bruce et al. of Hipsey et al)*

**Response:**   We added a references to these models along with the FLake model in the **Introduction**:

25   One-dimensional lake models, e.g. the FLake model (Mironov et al., 2010), DYRESM (Imberger and Patterson, 1981), GLM (Hipsey et al., 2019), play a major role in this process.

**Technical details**

*Line 8: capable OF*
Corrected.

*Line 98 : BEING used*
Corrected.

*Line 125: are -> is*
35   Corrected.

*Line 137 "into account" to the end of phrase*
Corrected.

*Line 147. Check sentence*
Reworked:

To calculate the dissolved carbon dioxide concentration in water, the same type of prognostic equation is used as for other gases.

*Line 155: unit is missing*
Added $m^{-1}$.

*Line 162: THE chose period*
Corrected.

*Line 165 capital T*
The sentence was removed.

*Line 173: comma after stage?*
Added.

*Line 185: how is density rho calculated (including solutes?)*

Solutes were not included in the current experiment (Alqueva is a freshwater lake), and the $\rho$ was calculated with the dependence on water temperature according to (McCutcheon, S. C., Martin, J. L., & Barnwell, T. O. (1993). *Water Quality. In Handbook of Hydrology* (pp. 11.11-11.73)).

*Line 191: comma after hours (possibly also before bigger)*

The sentences were reworked:

A linear interpolation was used to fill the gaps in data smaller than 3 hours. The gaps longer than 3 consecutive hours were substituted with the corresponding values from the land weather stations (Barbosa and Cid Almeida).

*Caption Fig. 4: include the information of 6-hours mean*
Added.

*Caption Fig. 4: errors-> temperature difference*
Corrected.

*Line 221: that -> which*
Corrected.

*Line 237. Remove "is"*
Removed.

*Line 251: covariAnce*
Corrected.

*Line 255: constant -> continuous*
Corrected.

*Line 266: which WAS SET TO A VALUE OF 8.48, which correspondED . . .*

5   Corrected.

*Figure 8: remove useless data between 10 and 17 March (or explain what can be seen)*
Removed. Figure 8 was remade.
*Line 274 comma after period*

10   Added.

*Line 270: good -> well*
Corrected.

[revised manuscript text omitted]

May 10, 2020

**1 Representation of biogeochemical processes in LAKE model**

**1.1 Governing equations for dissolved gases and organic carbon in a water column**

Evolution and vertical distribution of three dissolved gases are considered in the LAKE2.0 model, which are methane $CH_4$, oxygen $O_2$ and carbon dioxide $CO_2$. However, dissolved carbon dioxide is supposed to be always in carbonate equilibrium, so that it contributes to concentration of dissolved inorganic carbon (DIC), $C_{DIC} = C_{CO_2} + C_{HCO_3^-} + C_{CO_3^{2-}}$, and it is the change of DIC that reflects the number of carbon atoms in $CO_2$ molecules added to (or lost by) a solution from (to) atmosphere, bubbles, respiring organisms or decaying organical matter (see Section 1.2).

In addition, the content of dissolved organic carbon (DOC), particulate organic carbon (both living, POCL, and dead, POCD) are calculated. POCL includes carbon atoms contained in phytoplankton and zooplankton.

The species listed above obey the following equation system:

$$\frac{\partial C_{CH_4}}{\partial t} = \text{Dif}_A(C_{CH_4}) + B_{CH_4} - O_{CH_4}, \tag{1}$$

$$\frac{\partial C_{O_2}}{\partial t} = \text{Dif}_A(C_{O_2}) + B_{O_2} + P_{O_2} - R_{O_2} - D_{O_2} - S_{O_2} - O_{O_2}, \tag{2}$$

$$\frac{\partial C_{DIC}}{\partial t} = \text{Dif}_A(C_{DIC}) + B_{CO_2} - P_{CO_2} + R_{CO_2} + D_{CO_2} + S_{CO_2} + O_{CO_2}, \tag{3}$$

$$\frac{\partial \rho_{DOC}}{\partial t} = \text{Dif}(\rho_{DOC}) + E_{POCL} - D_{DOC}, \tag{4}$$

$$\frac{\partial \rho_{POCL}}{\partial t} = \text{Dif}(\rho_{POCL}) + P_{POCL} - R_{POCL} - E_{POCL} - D_{h,POCL}, \tag{5}$$

$$\frac{\partial \rho_{POCD}}{\partial t} = \text{Dif}(\rho_{POCD}) - \frac{\partial(w_g \rho_{POCD})}{\partial z} - D_{POCD} + D_{h,POCL}. \tag{6}$$

where $\text{Dif}_A(\bullet) \equiv \frac{1}{A}\frac{\partial}{\partial z}\left(A k_s \frac{\partial \bullet}{\partial z}\right)$, $\text{Dif}(\bullet) \equiv \frac{\partial}{\partial z}\left(k_s \frac{\partial \bullet}{\partial z}\right)$ are diffusion operators, $w_g$ is a sedimentation velocity of POCD particles. Equations (4)-(6) do not contain $A$, because they are not derived by horizontal averaging, but follow from assumption of horizontal homogeneity of respective biogeochemical variable. This is caused by uncertainty of estimating the flux of these substances at the sloping interface between water and sediments. The r.h.s of these equations represent diffusion (assuming $k_s = k_{s,t} + k_{s,m}$ with the same eddy diffusivity $k_{s,t}$ and molecular diffusivity $k_{s,m}$ for all species; molecular dissusivity is not included in POCL and POCD equations), sources and sinks due to the following processes:

- dissolution/exsolution of gases at the bubble-water interface ($B_{CH_4}$, $B_{O_2}$ and $B_{CO2}$);

- photosynthesis ($P_{O_2}$, $P_{CO_2}$, $P_{POCL}$);

- respiration ($R_{O_2}$, $R_{CO_2}$, $R_{POCL}$);

- biochemical oxygen demand in the water column ($D_{O_2}$, $D_{CO_2}$, $D_{DOC}$, $D_{POCD}$);

- sedimentary oxygen demand ($S_{O_2}$, $S_{CO_2}$);

- methane aerobic oxidation in the water column ($O_{CH_4}$, $O_{O_2}$, $O_{CO_2}$);

- death of living species ($D_{h,POCL}$)

All variables in the above list are positive definite, excepting $B_{CH_4}, B_{O_2}$ and $B_{CO2}$ that may be either positive or negative. All concentrations in (1)-(3) are expressed in mol/m$^3$ that allows for simple relations of sinks and sources in different equations based on stoichiometry of the respective reactions. Organic carbon variables DOC, POCL and POCD in (4)-(6) are molar concentrations of carbon atoms contained in these organic groups. Terms $B_{CO_2}, P_{CO_2}, R_{CO_2}, D_{CO_2}, S_{CO_2}, O_{CO_2}$ in (3) possess "CO$_2$" subscript because carbon atoms are supplied to or removed from DIC of a solution in a form of $CO_2$.

In the following, the parameterizations of processes related to O$_2$ and CO$_2$ dynamics are described, whereas formulations for CH$_4$ processes are presented in (Stepanenko et al., 2016).

The formulations for photosynthesis, respiration, biochemical oxygen demand and sedimentary oxygen demand basically adopted from (Stefan and Fang, 1994) and (Hanson et al., 2004).

**1.2 Carbonate equilibrium**

Carbonate equilibrium means the equilibrium in the following reactions:

$$CO_2 + H_2O \leftrightarrow H^+ + HCO_3^-, \tag{7}$$
$$HCO_3^- \leftrightarrow H^+ + CO_3^{2-}. \tag{8}$$

Involving kinetic constants of these reactions yields, that the DIC reads

$$C_{DIC} \equiv C_{CO_2} + C_{HCO_3^-} + C_{CO_3^{2-}} = C_{CO_2}\left[1 + k_1 10^{pH} + k_1 k_2 10^{2pH}\right]. \tag{9}$$

Here, the constants are given by Arrhenius equation:

$$k_i = k_{i0} \exp\left[-\frac{E_{act,i}}{R}\left(\frac{1}{T} - \frac{1}{T_0}\right)\right], \; i = 1, 2, \tag{10}$$

$R$ – universal gas constant, $k_1 = 4.3 * 10^{-7}$ mol/l, $k_2 = 4.7 * 10^{-11}$ mol/l, $E_{act,1} = 7.66 * 10^3$ J/mol, $E_{act,2} = 1.49 * 10^4$ J/mol. Thus, $C_{CO_2}$ is readily calculated given $C_{DIC}$ value, and vice versa, where $pH$ is an external parameter.

Carbon atoms are added or removed from carbonate equilibrium system in a form of $CO_2$ during respiration, photosynthesis and organic chemical

and physical processes, hence the change of $C_{DIC}$ equals to number of $CO_2$ consumed or produced. This explains the sense of terms in equation (3). For obtaining $CO_2$ flux across bubble surface or $CO_2$ diffusive flux to the atmosphere, $C_{CO_2}$ is needed and is calculated from (9).

**1.3 Boundary conditions for dissolved gases in a water column**

The top boundary condition (at the lake-atmosphere interface) for any dissolved gas concentration in the case of open water has the form:

$$\frac{k_s}{h}\frac{\partial C}{\partial \xi}\bigg|_{\xi=0} = F_C, \tag{11}$$

where $C$ is $C_{CH_4}, C_{O_2}$ or $C_{CO_2}$, and $F_C$ is the diffusive flux of a gas into the atmosphere, positive upwards. This flux is calculated according to the widely used parameterization:

$$F_C = k_{ge}(C|_{\xi=0} - C_{ae}), \tag{12}$$

with $C_{ae}$ being the concentration of the gas in water equilibrated with the atmospheric concentration and described by Henry law and $k_{ge}$, m/s, denoting the gas exchange coefficient, the so-called "piston velocity". The latter is written as:

$$k_{ge} = k_{600}\sqrt{\frac{600}{Sc(T)}}, \tag{13}$$

with the Schmidt number $Sc(T)$ having individual values for different gases and being temperature-dependent. The $k_{600}$ coefficient has been a subject of numerous studies, and a number concepts have been put forward to quantify it (Donelan and Wanninkhof, 2002). The proper computation of this coefficient should account for the effects of a number of factors such as turbulence in adjacent layers of water and air, wave development and breaking, cool skin dynamics. The surface renewal model (MacIntyre et al., 2010; Heiskanen et al., 2014), used in LAKE2.0 model, "integrates" those effects through the near-surface dissipation rate of turbulent kinetic energy:

$$k_{600} = \frac{C_{1,SR}(\epsilon|_{\xi=0}\nu_w)^{\frac{1}{4}}}{\sqrt{600}}, \tag{14}$$

where $\nu_w$ designates molecular viscosity of water, $C_{1,SR} = 0.5$ is an empirical constant. TKE dissipation rate is available directly from $k - \epsilon$ closure.

When a lake is covered by ice, $F_C = 0$, which neglects contribution of diffusion through ice cracks.

**1.4   Photosynthesis**

The intensity of photosynthesis in terms of oxygen molecules production is expressed as:

$$P_{O_2} = \frac{P_{max} L_{min} \rho_{Chl-a}}{H_{sec} \mu_{O_2}}. \tag{15}$$

The denominator here serves co convert units in the r.h.s. from mg/(l*h) to mol/(m$^3$ s). The $P_{max}$ value expresses limitation of oxygen production by temperature in a form:

$$P_{max} = C_P \theta_P^{(T-T_0)}, \tag{16}$$

so that $C_P$ is a value of $P_{max}$ at the reference temperature $T = T_0$. The limitation of oxygen production by the available photosynthetically active radiation PAR ($S_{PAR}$) is given by the Haldane kinetics:

$$L_{min} = \frac{S_{PAR}(1 + 2\sqrt{C_{Lmin,1}/C_{Lmin,2}})}{S_{PAR} + C_{Lmin,1} + S_{PAR}^2/C_{Lmin,2}}. \tag{17}$$

The PAR intensity delivering maximum to a limiter $L_{min}$ (=1) is $S_{PAR} = \sqrt{C_{Lmin,1}C_{Lmin,2}}$. In the model, these coefficients are specified as (Stefan and Fang, 1994; Megard et al., 1984):

$$C_{Lmin,1} = C_{PAR} \theta_{PAR}^{(T-T_0)}, \tag{18}$$
$$C_{Lmin,2} = \mathrm{H}(T - T_{00})C_{Lmin,2,>T_{00}} + [1 - \mathrm{H}(T - T_{00})]C_{Lmin,2,<T_{00}}, \tag{19}$$

with H($\bullet$) denoting a Heavyside function, and $T_{00}$ standing for another reference temperature. It is seen from (17), that $L_{min} \to 0$ if $S_{PAR} \to 0$ and $S_{PAR} \to \infty$, i.e. PAR ihnibits photosynthesis at both low and high values of its intensity. The PAR instensity $S_{PAR}$ is expressed in a number of photons per square meter per hour, so that:

$$S_{PAR} = H_{sec} T_{J \to Eins} S_{PAR}^*, \tag{20}$$

where $H_{sec} = 3600$ s and $S^*_{PAR}$ is PAR intensity in W/m². The coefficient transforming from J to Einstein (Einstein is an energy of Avogadro number of photons), $T_{J \to Eins}$, is estimated assuming the uniform distribution of energy in PAR region, which yields:

$$T_{J \to Eins} = \frac{\lambda_{PAR}}{N_A h_P c}, \tag{21}$$

with $N_A, h_P, c$ denoting Avogadro number, Planck constant and the light speed in vacuum, respectively, all in SI units.

The treatment of chlorophyll-a concentration $\rho_{Chl-a}$ is given in Section 1.10.

Finally, from the gross photosynthesis reaction:

$$6CO_2 + 12H_2O + photons \to C_6H_{12}O_6 + 6O_2 + 6H_2O, \tag{22}$$

or, in a shortened form:

$$CO_2 + 2H_2O + photons \to CH_2O + O_2 + H_2O, \tag{23}$$

we see that the carbon dioxide consumption equals oxygen production, i.e. $P_{CO_2} = P_{O_2}$.

Equation (22) also implies that $P_{POCL} = P_{CO_2}$.

**1.5 Respiration**

P.Hanson et al. (Hanson et al., 2004) assume, that respiration is performed by "living particles", i.e. POCL, only in epilimnion, and may be scaled by gross primary production (i.e., photosynthesis rate), $R_{POCL} = \alpha_{POCL} P_{POCL}$, $\alpha_{POCL} = 0.8$. In contrast, we assume that this process happens at all depths where enough oxygen *in situ* to be used in respiration is available, with the same scaling. Evidently,

$$R_{O_2} = R_{CO_2} = \alpha_{POCL} P_{POCL}. \tag{24}$$

**1.6 Biochemical oxygen demand (BOD)**

We treat biochemical oxygen demand as a consumption of oxygen during degradation of dead organic particles (POCD) $D_{POCD}$ and dissolved organic

Table 1: Constants in photosynthesis model

| Constant | Units | Value |
|---|---|---|
| $C_P$ | h$^{-1}$ | 9.6 |
| $\theta_P$ | n/d | 1.036 |
| $T_0$ | °C | 20 |
| $T_{00}$ | °C | 10 |
| $\mu_{O_2}$ | g/mol | 32 |
| $H_{sec}$ | s | 3600 |
| $C_{PAR}$ | Einstein/(m$^2$*h) | 0.687 |
| $\theta_{PAR}$ | n/d | 1.086 |
| $C_{Lmin,2,>T_{00}}$ | Einstein/(m$^2$*h) | 15. |
| $C_{Lmin,2,<T_{00}}$ | Einstein/(m$^2$*h) | 5. |
| $\lambda_{PAR}$ | m | $5.5 * 10^{-7}$ (550 nm) |

carbon (DOC) $D_{DOC}$, following (Hanson et al., 2004); they suggest that $D_{POCD} = \rho_{POCD}/\tau_{POCD}$, $D_{DOC} = \rho_{DOC}/\tau_{DOC}$ with time scales $\tau_{POCD} = 20D_{sec}$, $\tau_{DOC} = 200D_{sec}$ ($D_{sec}$ is a number of seconds in a day). Thus, the BOD rate is:

$$D_{O_2} = D_{CO_2} = \left( \frac{\rho_{POCD}}{\tau_{POCD}} + \frac{\rho_{DOC}}{\tau_{DOC}} \right). \tag{25}$$

**1.7 Sedimentary oxygen demand (SOD)**

The sedimentary oxygen demand appears as a sink in (2) and in essence is the contribution of the vertical flux of O$_2$ at the lake's bottom to the horizontally averaged oxygen concentration:

$$S_{O_2} = -\frac{F_{SOD}}{A} \frac{\partial A}{\partial z}. \tag{26}$$

Basing on the argument that SOD is controlled by both diffusion (governed by Fickian law) and biochemical consumption (described by Michaelis-Menthen kinetics), (Walker and Snodgrass, 1986) derive:

$$F_{SOD} = \mu_\beta \frac{C_{O_2}}{k_{O_2,SOD} + C_{O_2}} + k_c C_{O_2}, \tag{27}$$

where $\mu_\beta$ is proportional to organics oxidation potential rate in sediments, and $k_c$ is the mass transfer coefficient. Both are thought to be exponentially dependent on temperature:

$$\mu_\beta = \mu_{\beta,0}\theta_{\mu_\beta}^{T-T_{\mu_\beta}}, \, k_c = k_{c,0}\theta_{k_c}^{T-T_{k_c}}. \tag{28}$$

The stoichiometry of SOD is assumed to be close to that of BOD (**??**), therefore, $S_{CO_2} = S_{O_2}$. Additionally, the flux of $O_2$ due to SOD at the lake bottom, $F_{SOD}$, is used as the bottom (lake deepest point) boundary condition for the oxygen equation (2).

Table 2: Constants in sedimentary oxygen demand model

| Constant | Units | Value |
|---|---|---|
| $\theta_{\mu_\beta}$ | n/d | 1.085 |
| $\theta_{k_c}$ | n/d | 1.103 |
| $T_{\mu_\beta}$ | K | 25 |
| $T_{k_c}$ | K | 20 |
| $\mu_{\beta,0}$ | mol/(m²*s) | $0.5/(\mu_{O_2}D_{sec})$, $[\mu_{O_2}] = g/mol$ |
| $k_{c,0}$ | m/s | $0.045/D_{sec}$ |

**1.8 Exudates and death rate of POCL**

Hanson et al. suggest exudation to be scaled with photosynthesis rate, $E_{POCL} = \beta_{POCL}P_{POCL}$, $\beta_{POCL} = 0.03$ and the death rate to be defined as $D_{h,POCL} = \frac{\rho_{POCL}}{\tau_{Dh}}$, where time scale $\tau_{Dh}$ ranges from $1.1D_{sec}$ in hypolimnion to $33D_{sec}$ in epilimnion.

**1.9 Sedimentation of organic particles**

In the current model version we use the Stokes sedimentation velocity below the mixed layer:

$$w_s = \frac{4}{3A}\frac{\Delta g d^2}{\nu_m}, \tag{29}$$

and the high-Reynolds-number limit of this variable

$$w_s = \sqrt{\frac{4}{3B}\Delta g d} \tag{30}$$

in the mixed layer. Here, $\Delta = \rho_p/\rho_{w0} - 1$, $\rho_p$ is a particle's density, and $d$ – its diameter, the typical values for constants may be chosen as $A = 30.0$, and $B = 1.25$ (Song et al., 2008), and the density of organic particles as $1.25 \ g/cm^3$ (Avnimelech et al., 2001).

**1.10 Chlorophyll-a dynamics**

The chlorophyll-a dynamics in the model follows a simple scheme suggested in (Stefan and Fang, 1994), where chlorophyll-a density is calculated as:

$$\rho_{Chl-a} = \rho_{Chl-a,0}H(H_a - z), \tag{31}$$

where the active layer, $H_a$, is a maximum value between mixed-layer depth, $H_{ML}$, and the photic zone depth, $H_{PZ}$. The mixed-layer depth is defined as the depth of maximum Brunt-Väisälä frequency, and the photic zone depth is estimated as the depth at which the PAR irradiance drops to 10% of its surface value. The mean chlorophyll-a concentration in the active layer, $\rho_{Chl-a,0}$, is assigned according to a trophic status of the lake: $2*10^{-3}$ mg/l for oligotrophic lakes, $6*10^{-3}$ mg/l for mesotrophic lakes and $15*10^{-3}$ mg/l for eutrophic lakes. In turn, the trophic status is formally defined from the water turbidity. The Secchi disk values of 2 m and 3.5 m are used to distinguish between eutrophic and mesotrophic, mesotrophic and oligotrophic states, respectively. These thresholds are expressed in the model through light extinction coefficient values, $\alpha$, using Poole and Atkins formula (Poole and Atkins, 2009):

$$\alpha = \frac{k_{PA}}{z_{SD}}, \tag{32}$$

where $z_{SD}$ is the Secchi disk depth and $k_{PA} = 1.7$. The above chlorophyll-a scheme is identical to that of (Stefan and Fang, 1994), excepting for it does not take into account the annual cycle of $\rho_{Chl-a,0}$.

**2 Sensitivity tests**

[Figure]

Figure S1: ML water temperature, original and modified model results and their errors when compared to the observations.

[Figure]

Figure S2: DO concentration at different chlorofill-a concentration values.

[Figure]

Figure S3: $CO_2$ concentration at different chlorofill-a concentration values.

[Figure]

Figure S4: Dissolved $CO_2$ concentrations at different pH values.

[Figure]

Figure S5: ML temperature at different hypolimnetic diffusivity coefficient values.

---

## Author Comment (AC2) · 11 May 2020

**Numerical study of the seasonal thermal and gas regimes of the large artificial reservoir in Western Europe using LAKE2.0 model**

Authors' responses to the Referee #2 comments

Maksim Iakunin[1], Victor Stepanenko[2] , Rui Salgado[1] , Miguel Potes[1] , Alexandra Penha[3,4] , Maria Helena Novais[3,4] , and Gonçalo Rodrigues[1]

*miakunin@uevora.pt*

[1]Department of Physics, ICT, Institute of Earth Sciences, University of Évora, 7000 Évora, Portugal
[2]Lomonosov Moscow State University, GSP-1, 119234, Leninskie Gory, 1, bld. 4, Moscow, Russia
[3]Water Laboratory, University of Évora, P.I.T.E. Rua da Barba Rala No1, 7005-345 Évora, Portugal
[4]Institute of Earth Sciences — ICT, University of Évora, Rua Romão Ramalho 59, 7000-671 Évora, Portugal

**Contents**

**Introduction. Document structure**

This document contains authors' responses to the comments of the Anonymous Referee. The document structure is the following:

- Referee's comments are numbered and given in *italic font*. General, specific, and technical comments come separately.

- Authors' response follows the comment and starts after **"Response:"** with normal font.

- The text from the article itself (if some changes are done, and if it is reasonable to provide it) is typed with `typewriter font` and separated from the response with an extra blank line.

- *Technical comments and mistakes* are not numbered, and authors' response follows immediately.

Reviewed manuscript with all the corrections is given after all responses. It contains the changes and proposals of **two** Referees and was prepared using LaTeXdiff package for better understanding of what has been changed.

**Anonymous Referee #2**

*General comments*

*It is useful that this paper presents a model comparison that focuses on the factors that would be most important in influencing the heat and gas fluxes from a lake. It was good to see model comparisons focusing on the mixed layer. However, there is a need to clearly state the criteria used to define the mixed layer. I was pleased to see the specific comparison of measured heat fluxes and gas fluxes collected at high resolution with simulated data from two models. This not commonly done and is a unique and valuable aspect of this paper. And for these reasons I think this paper does document important progress in lake model development and does deserve to be accepted for publication following revision.*

*I think for a modeling study such as this there is a need for more information on calibration. How the model was calibrated and what the final error levels were. It doesn't need to be extensive but as a minimum I would like to see a listing of the final calibrated parameter values, as well as a brief description of what each parameter does. A scatter plot of the simulated vs measured temperature. And some statistics on model fit (ie RMSE , MAE etc ) Furthermore, I'm assuming that the model was calibrated against measurements of water temperature, but this may not entirely be the case since there were measurements of gas concentrations and heat fluxes that could in theory also be used for calibration. What was used for calibration should be clearly stated.*

*One of the most important aspects of this paper is the comparison between the simulated gas fluxes with measured data. Therefore, I do think there is a need to better describe the equations governing $CO_2$ and $O_2$ concentrations. I was not that familiar with Lake 2.0 but after searching a bit I found that this is not the first time $CO_2$ and $O_2$ have been simulated with Lake 2.0, even though this paper may be one of the best verification studies. I would like to see some overview description of the main processes affecting the $CO_2$ and $O_2$ concentrations and also more clear references to the original publications where the equations describing these processes are completely defined.*

*There were two things that were changed in the version of the model used in this study (the fixed pH value and the equations affecting diffusion in the hypolimnion) and also one assumption (fixed chlorophyll concentration) which I suspect and which later in the paper the authors also suspect leads to errors in the simulated oxygen concentration. I think all three of these should be evaluated in a sensitivity analysis as part of the paper in section 2.4 as was done with the light extinction coefficient.*

*There are quite a few small language errors in the paper. I have tried to suggest solutions to many in the technical comments. These should not be allowed to take away from the good scientific and technical aspects of this study, so I think it would be good to have paper carefully proofread for language before the final submission.*

**Response:** We thank the Reviewer for the positive comments and thorough revision of the manuscript. The paper was edited very carefully and modifications and improvements were made. A Supplementary file was created as well, where more details of LAKE2.0 biogeochemistry processes are described and figures of sensitivity tests are shown. Below, we address every comment and explain the corresponding changes.

**Specific comments**

**Comment 1**

*Abstract - Since you mention the Flake model in the abstract I think you should have a brief statement about how well it worked compared to Lake2.0.*

**Response:** The end of the abstract was changed to:

The results demonstrated that both models well captured the seasonal variations in water surface temperature and the internal thermal structure of the Alqueva. The LAKE2.0 model showed slightly better results and satisfactorily captured the seasonal gas regime.

**Comment 2**

*Line 40 - what do you mean by "to complete the results". Does Flake do something that Lake2.0 does not? Or are you comparing the results of the two models*

**Response:** FLake does not do anything more than LAKE2.0, and by "complete" we meant to use its results as a supplement. As we, however, compared the results of two models, the sentence was rewritten:

...FLake model was used as a reference to compare the results of thermodynamic characteristics of the reservoir.

**Comment 3**

*Lines 98-99 I think you could give a bit more information. What type of errors? How much missing information was there? Linear interpolation?*

**Response:**  In this particular place of the text by errors we meant missed values. They occurred during the equipment maintenance (short gaps that were filled with linear interpolation) or when the equipment was temporarily dismounted. In the last case, when the gaps in data were larger than 3 hours, data from nearby stations were used instead. For instance, in case of lack in downward radiation data measured at the floating platform we used the values from the weather station on the shore. The corresponding sentence was reworked:

> Missed data (gaps in data smaller than 3 hours) were carefully filled using linear interpolation. Longer gaps were substituted with values from closest weather stations.

**Comment 4**

*Line 104 Are not these fluxes also occurring through the surface?*

**Response:**  They are. The sentence was corrected:

> ...through a sloping bottom and water-atmosphere surfaces.

**Comment 5**

*Line 105 and unlike Hostetler model I don't know what you mean by unlike Hostetler model. Are you using this model as well? Or our components of this model embedded in Lake2.0?*

**Response:**  No, none of the components of Hostetler model is used in the LAKE2.0 and this model did not participate in the experiment. The reference to it was removed from the sentence:

> Water temperature profile is simulated explicitly in LAKE2.0 and a number of biogeochemical processes is represented, which makes it capable to reproduce the transfer of $CO_2$ and $CH_4$ from and to the atmosphere.

**Comment 6**

*Line 141 the description of photosynthesis is rather unusual. Is it really reasonable to assume that chlorophyll remains constant while photosynthesis is changing? Perhaps this simplification can be justified by the fact that the modeling is mainly looking at gas exchange and not the biology of the lake. However, photosynthesis will affect both O2 and CO2. Assuming a constant chlorophyll concentration could greatly under or over estimate the total photosynthesis in the epilimnion. I think there should be more justification for the constant chlorophyll assumption. Perhaps a sensitivity analysis on how changes in chlorophyll affect the gas flux estimates.*

**Response:**  The model uses "pre-defined" values of chlorophyll-a concentration (three options) depending on PAR attenuation coefficient. Assuming this, chlorophyll-a concentration remained constant during almost the whole simulation. However, do avoid any confusion, the sentence on the constant chlorophyll value was removed from the text and the part describing the calculation of the chlorophyll-a from PAR attenuation coefficient was added. We also added a Supplementary material to the article where the model biogeochemical processes are widely described and results of sensitivity tests for pH chlorophyll-a concentration, hypolimnetic diffusivity coefficient, and

PAR attenuation coefficient are provided.

Photosynthesis is given by Haldane kinetics where chlorophyll-a concentration in mixed layer is computed from photosynthetic radiation extinction coefficient (Stefan and Fang,
5  1994), and assumed zero below.

**Comment 7**

*Later in lines 150-160 you document a large seasonal variation in the attenuation coefficient. How much of this is due to changes in chlorophyll? Could this variability invalidate the assumption of a constant chlorophyll concentration? Also in this section the model was modified to allow the*
10  *input of a varying extinction coefficient which is a good idea. However, this could be described more clearly. It is stated that "introduce a new variable,the water extinction coefficient for photo- synthetically active radiation (PAR), to the model setup" How does this coefficient differ from the coefficients described in line 120. Perhaps you mean that the existing coefficient described in line 120 was changed from a fixed model coefficient to a time varying one? However Im still a little*
15  *confused since PAR is usually considered to be between 400-700 nm and measured as a photon flux density, whereas I would think that the coefficient described on line 120 would have a wider bandwidth and would be measured in terms of watts*

**Response:**   Attenuation coefficient depends mainly on water turbidity and consequently depends on suspended and dissolved material in water. Higher chlorophyll concentration can indirectly in-
20  crease the water turbidity and attenuation coefficient. The assumption of a constant chlorophyll concentration is set to mixed layer. The measurements of the attenuation coefficient were done in the layer from surface to 3 meter depth. In table 4 chlorophyll concentrations are presented for surface, 1, 2 and 3 m depth and is noticed that, despite in the September 2018 bloom, the values are almost constant in these depths. Thus, the seasonal variability of attenuation co-
25  efficient, accounted in this study through the measurements campaign, is an asset to compute reliable chlorophyll concentrations even if this concentration is set constant in the mixed layer. Supplementary material has a description of the processes that are included in the model.
   There are 4 bands of the shortwave radiation tn the LAKE2.0 model, and each has it's own attenuation coefficient (Table 1). Only the coefficient for PAR range was added as an input variable (the rest remained constants). Corresponding changes were done in the text.

| UV | 3.13 |
| --- | --- |
| PAR | *variable* |
| NIR | 1.73 |
| IR | 1.087E+3 |

Table 1: Attenuation coefficients in LAKE2.0.

   Attenuation coefficient units are $m^{-1}$ while the solar energy was measured in terms of $Wm^{-2}$ indeed. Conversion of PAR intensity from $Wm^{-2}$ to $N_{photons}\ m^{-2}\ hour^{-1}$ could be done with the equation (20) and (21) provided in the Supplementary material.

**Comment 8**

35  *Lines 165-166 need to be made clearer.*

**Response:**   The sentence was removed from the paragraph.

**Comment 9**

⁵ *Starting at line 177 there are two changes to the model described one concerning pH and the other concerning hypolimnetic diffusivity. Sensitivity analysis should be done for both of these and these results would be better presented in the section starting on line 150*

**Response:**   The paragraph concerning diffusion hypolimnion was moved to the Section 2.4. Figures for the model sensitivity tests for this coefficient as well as for pH and chlorophyll-a concentration values were added to the Supplementary materials.

**Comment 10**

¹⁰ *Line 191 this information is better place in the section on observational data (see comment above).*

**Response:**   Corresponding changes were done (see response to Comment 3).

**Comment 11**

¹⁵ *Line 207 Model simulations of the mixed layer depth (MLD) are discussed. The method for defining the MLD should be described in the methods section. Also in figure 4 it would be good to show a plot of the variations in the MLD over time. In the caption of fig 4 describe what the dashed horizontal lines represent.*

**Response:**   A new paragraph with a discussion on ML depth results was added to the end of Section ²⁰ **3.1 Water temperature.** A method for the estimation of ML depth is defined there and the results are shown. A new figure with time series of the evolution of ML depth is introduced as well.

The other important parameter which is essentially connected with lake vertical thermal structure is depth of mixed layer. To estimate it we assumed that ML ends at a point of half ²⁵ of the maximum temperature gradient (but not less than 0.5 °C). Such criterion was used for observed data and LAKE2.0 results. In Flake, the ML depth is a major diagnostic variable, updated each time step using a sophisticated formulation, that treats both convective and stable regimes (see Mironov (2010)). Time series of the ML depth for the 2017 and 2018 Alqueva's stratification periods are shown in Fig. 6. Curves of ML depth calculated from ³⁰ measurements and LAKE2.0 results coincide quite well. However, since the simulated water temperature profiles are more smooth, LAKE2.0 ML depth has more "downward" peaks in the figure. Although FLake tends to underestimate ML depth, the general pattern of it correlates with measurements.

³⁵ Caption for the Fig. 4 was corrected as well.

[Figure]

**Comment 12**

*Line 232 states "water temperature of thermocline beneath the ML at any depth" Don't you mean water temperature of the hypolimnion?*

**Response:**   That is right in general, however, particularly here we were speaking about the FLake model outputs. FLake does not represent meta- and hypolimnion, instead it has mixed layer and thermocline below it (a layer of water with negative temperature gradient down to the bottom). In FLake, this thermocline is described as parametrized curve which form is similar for every lake (this concept of self-similarity is the essence of this model). Depth-normalized curve is defined by the shape factor parameter, thus, if one knows it plus temperatures on the top and bottom of the thermocline, it is possible to compute temperature at any depth of this curve. The sentence was reworked:

    FLake provides ML depth, shape factor for the thermocline curve, ML and bottom tem-
    perature. Using these values it is possible to access water temperature of thermocline at any
    depth.

**Comment 13**

*Line 252 states "Present results show comparable differences between the FLake and the LAKE2.0 models and EC measurements over lakes (Stepanenko et al., 2014; Heiskanen et al., 2015)" To me present studies means the study being described in this paper. Do you mean something like other recent studies?*

**Response:**   This is a mistake in the text. Now `Present results show ...` is changed to `Recent works showed ...`

**Comment 14**

*Lines 255-259 Is it possible to estimate the depth of the horizontal flows from the surface temperature of the inflowing river? It seems like this would be most significant if the inflows are*

*moving through the surface layer.*

**Response:**   The LAKE2.0 model is capable to include the effects of inflow/outflow rivers, however, we did not include this option during the experiment: firstly, there are no regular measurements on the tributary rivers, secondly, we assumed that their effect is not significant. Our assumption was based no the following simple estimation.

We can write the thermal balance of the mixed layer in the following form:

$$\rho c V \frac{\partial T_{ML}}{\partial t} = q \rho c (T_{river} - T_{ML}) + S_{lake}(-H - LE + radiation),$$

where $\rho$ is water density, $c$ — water heat capacity, $V$ — volume of the ML, $T_{ML}$ and $T_{river}$ are ML and river temperature, respectively, $q$ is the inflow rate, $m^3 s^{-1}$, $S_{lake}$ — area of the lake surface, $H, LE, radiation$ — sensible, latent heat and radiation fluxes. The part $q \rho c (T_{river} - T_{ML})$ shows the amount of heat brought by the river.

| | | |
|---|---|---|
| $q$ | $36\ m^3\ s^{-1}$ | provided by EDIA for April 2020 |
| $\rho$ | $999\ kg\ m^{-3}$ | — |
| $T_{ML}$ | $16.5\ °C$ | average of the measurements in April 2020 |
| $T_{river}$ | $17.5\ °C$ | no data originally, assumed as an average daytime air temperature |
| $c$ | $4185\ J\ kg^{-1}\ K^{-1}$ | — |
| $S_{lake}$ | $2.5{\times}10^8\ m^2$ | — |
| $H$ | $30\ W\ m^{-2}$ | average value for April |
| $LE$ | $80\ W\ m^{-2}$ | „ |

Table 2: Example of values for the heat capacity estimation.

Using corresponding values given in Table 2, we compared $q \rho c (T_{river} - T_{ML})$ with $S_{lake} H$ and $S_{lake} LE$ and found out that the heat brought by the river tributary is only about 2% of heat lake loses from sensible heat and 0.8% of latent heat.

**Comment 15**

*Line 270 In the second week of May, CO2 probe accidentally dismounted from the platform and remained. - I think you should just remove these data from the plot You clearly do not believe they are meaningful and have a good explanation for this.*

**Response:**   Figure 8 was remade and Fig. 9 was combined with Fig. 8. Corresponding changes have been made in the paragraph.

**Comment 16**

*Lines 298-299 Chlorophyll concentrations are given in mg/l should these be ug/l (10-6g/l)? The mg/l concentrations that are given are very high and would be considered typical of a highly eutrophic lake. They would also certainly greatly affect the O2 concentration I have the same concern for the values in table 4.*

**Response:**   A mistake was made cholorophyll-a units: it mg m$^{-3}$. Corrections were made in text and in the Table 4.

[Figure]

Figure 1: Result figure discussed in Comment 15

**Comment 17**

*Line 322 You state "Such errors could be due to the sporadic input of measured wind speed, which values change rapidly" First I would suggest sporadic nature rather than input. But I also think this needs more of an explanation Are we talking about errors in both latent and sensible heat? And what is the mechanism by which sporadic winds are increasing model error?*

**Response:**   No, here we were speaking only about latent heat. We noticed that latent heat in the model is sensitive to strong increase of wind speed. In such cases, a local peak of latent heat in the model may occur, which usually does not coincide with the measurements. Corresponding sentence was rephrased to become clearer:

    Such errors occur mainly in periods when the wind increases suddenly. Strong single high hourly wind input data cause high latent heat simulated values, not always confirmed by the observations.

**Comment 18**

*Line 325 You state "On the second year of the experiment (October 2018, when the probe was returned to the platform), simulated CO2 values did not show big errors despite the fact that pH value remained constant during the whole simulation period." But was not the pH also constant during the first year? Were there larger errors in the first year due to the fixed pH?*

**Response:**   Yes, the value of pH remained constant during the whole simulation. Here we wanted to point out that, despite that fact, the simulated values of $CO_2$ have not "gone far" from the observations after 18 months of the simulation, in the end of October 2018. The corresponding

sentence was rephrased:

On the 18th month of the experiment (October 2018, when the probe was returned to the platform), the simulated $CO_2$ values did not show large residuals despite the fact that pH value remained constant during the whole simulation.

*Comment 19*

*Line 334 This final paragraph needs to be reworked First I think you should be stressing that the Lake 2.0 model was shown to accurately simulate the heat fluxs and gas fluxes from the ML. I think this is one of the major model developments being described here. Secondly, I don't think you should start out by say that Flake is good model — Im sure this is true but it is not the purpose of this paper. You should be stating that Lake 2.0 is as good or better than Flake as you have shown in some of the comparisons in the paper. Finally in terms of using these two models to improve weather predictions you state that Flake has lower computational demands. By why not give some numbers on this? How much slower is Lake 2.0? Is it realistic to think it could be used to support weather prediction in the future?*

**Response:**   The final paragraph was reworked. We removed the part about computational performance. FLake model is used in several NWP systems, thanks to its simplicity and satisfactory representation of the evolution of the lake surface temperature, which is the most important parameter from the point of view of weather forecasting and, therefore, support weather prediction. We do not state that LAKE2.0 would support it in future, but it can be used in many tasks in fields of ecology, biogeochemistry, etc.

Performed simulations showed that LAKE2.0 model accurately simulates the lake thermal regime and the heat and gas fluxes from the ML. In terms of water temperature profile, LAKE2.0 demonstrated better performance than the FLake model. The results are encouraging as to the ability of the LAKE2.0 model to represent the evolution of physicochemical profiles inside the lakes, and may be used operationally in the future, coupled with weather prediction models, to forecast variables useful in the management of water quality and aquatic ecosystems. Similarly, the results indicate that the LAKE2.0 model could be used in climate modelling to estimate the impacts of the climate change on the thermal and gas regimes of the lake.

**Technical details**

*Line 1 (Suggested change) The Alqueva reservoir (southeast of Portugal) being the largest artificial lake in Western Europe and a strategic freshwater supply in the region. The reservoir is of scientific interest and monitored in order to maintaining the quality and quantity of water and evaluate its impact on the regional climate. To support these tasks we conducted numerical studies of the thermal and gas regimes in the lake.*

Suggestions were considered and the sentence was corrected.

*(Suggested change)supplemented by the data observed at the weather stations and the floating platforms deployed during the field campaign of the ALOP (ALentejo Observation and Prediction System) project. One-dimensional model LAKE2.0 was used for the numerical studies.*

5      Weather stations and floating platform were deployed and established before the ALOP. Other corrections were done.

*line 8 this parameterization » this model?*
Corrected.

*Line 14 particpants » regulators*
Corrected.

*Line 20 allow to use them » allows them to be used*
15   Corrected.

*Line 25 models are important models is important*
Corrected.

20      *Line 30 for the lake ecosystem vital activity » regulating lake ecosystem processes*
Corrected.

*Line 39 allowing to reproduce the concentrations » that simulates the concentrations*
Corrected.

*Line 43 spelling (hypolimnion) forced with the observed data » forced with the observed meteorological data*
Corrected.

30      *Line 49 spreading along 83 km over » spreading over 83 km of of Guadiana » of the Guadiana*
Corrected.

*Line 51 the capacity of water » the storage capacity of water*
Corrected.

*Line 59 in favourable position.  » into a favourable position.  Rainfall seasons normally last from » Seasonal rainfall normally occurs between*
Corrected.

40      *Lines 64-65 Geographical and climatological factors make the Alqueva reservoir a vital source of fresh water needed to support the population and economy in the region, while on the other hand, an increasing anthropogenic*
Corrected.

45      *Line 71 air columns, over the water-atmosphere interface, and in the shores » air columns, at the water-atmosphere interface, and on the shores*
Corrected.

Line 73 4 floating platforms » four floating platforms
Corrected.

Line 76 was settled on the platform » was deployed on the platform
Corrected.

Line 77 an eddy-covariance system, Campbell Scientific Irgason, provides data for atmospheric
» an eddy-covariance system, Campbell Scientific provides data of atmospheric
Corrected.

Line 89 and for punctually vertical profiles » and was occasionally used to collect vertical profiles
Corrected.

Line 94 were obtained in automatic regime and transferred » were automatically downloaded and transferred
Corrected.

Line 96 weather stations and conduct measurements, to collect » weather stations, to conduct more detailed measurements, and to collect
Corrected.

Line 125 condition »conditions
Corrected.

Line 126 condition is»conditions are
Corrected.

Line 161 (Suggested change) In addition to LAKE2.0, The FLake model was used to simulate water temperature for the chosen period. FLake model (Mironov,
Corrected.

Line 162 a two-layer representation of the temperature profile » a two-layer representation of the lake's thermal structure
Corrected.

Line 196 (Suggested change) Water temperature is a crucial factor for Numerical Weather Prediction (NWP) applications, and as a regulator of lake ecosystem activity, and their ecosystems.
Corrected.

Line 237 its integral energy » the simulated heat content of the entire water column.
Corrected.

Line 245 is capable to calculate » are capable of calculating
Corrected.

*Line 246 and the profile 7 represents » and figure?? shows.*
Corrected.

*Line 270 are represented quite good » are represented quite well*
Corrected.

*Line 292 In November the turnover happens » In November, following turnover*
Corrected.

*Line 312 Alqueva reservoir with using the LAKE2.0 » Alqueva reservoir using the LAKE2.0*
Corrected.

*Line 316 correlation coefficients are 0.99 for both » correlation coefficients for the relationship between simulated and measured temperature are 0.99 for both*
Corrected.

*Line 317 FLake shows overestimation about 1.5 » FLake shows an overestimation of about 1.5*
Corrected.

*Line 318 show the same rate of overestimation » show the same level of overestimation*
Corrected.

*Line 324 good accordance » good corrospondance.*
Corrected.

*Line 328 of modernisation of LAKE2.0 » inclusion of a more complete description of the process regulating photosynthesis and respirations in the LAKE2.0 model*
Corrected.

*Line 329 (Suggested change) Although measured oxygen concentrations are well simulated values of O2 over short time intervals, the annual Alqueva oxygen cycle cannot be reproduced because the model does not respond to changes in algal concentration (underestimation of O2 values) and winter minimum (high overestimation). Winter overestimation is supposedly due to the relatively low water temperatures. Why above is it supposedly? Are you not sure?*
Corrected. We say "supposedly" because it remains the principal hypothesis for explanation of such big residuals. The depth of the reservoir of the used bathymetry might be the cause as well, either everything in place to a certain extent.

[revised manuscript text omitted]

May 10, 2020

**1 Representation of biogeochemical processes in LAKE model**

**1.1 Governing equations for dissolved gases and organic carbon in a water column**

Evolution and vertical distribution of three dissolved gases are considered in the LAKE2.0 model, which are methane $CH_4$, oxygen $O_2$ and carbon dioxide $CO_2$. However, dissolved carbon dioxide is supposed to be always in carbonate equilibrium, so that it contributes to concentration of dissolved inorganic carbon (DIC), $C_{DIC} = C_{CO_2} + C_{HCO_3^-} + C_{CO_3^{2-}}$, and it is the change of DIC that reflects the number of carbon atoms in $CO_2$ molecules added to (or lost by) a solution from (to) atmosphere, bubbles, respiring organisms or decaying organical matter (see Section 1.2).

In addition, the content of dissolved organic carbon (DOC), particulate organic carbon (both living, POCL, and dead, POCD) are calculated. POCL includes carbon atoms contained in phytoplankton and zooplankton.

The species listed above obey the following equation system:

$$\frac{\partial C_{CH_4}}{\partial t} = \text{Dif}_A(C_{CH_4}) + B_{CH_4} - O_{CH_4}, \tag{1}$$

$$\frac{\partial C_{O_2}}{\partial t} = \text{Dif}_A(C_{O_2}) + B_{O_2} + P_{O_2} - R_{O_2} - D_{O_2} - S_{O_2} - O_{O_2}, \tag{2}$$

$$\frac{\partial C_{DIC}}{\partial t} = \text{Dif}_A(C_{DIC}) + B_{CO_2} - P_{CO_2} + R_{CO_2} + D_{CO_2} + S_{CO_2} + O_{CO_2}, \tag{3}$$

$$\frac{\partial \rho_{DOC}}{\partial t} = \text{Dif}(\rho_{DOC}) + E_{POCL} - D_{DOC}, \tag{4}$$

$$\frac{\partial \rho_{POCL}}{\partial t} = \text{Dif}(\rho_{POCL}) + P_{POCL} - R_{POCL} - E_{POCL} - D_{h,POCL}, \tag{5}$$

$$\frac{\partial \rho_{POCD}}{\partial t} = \text{Dif}(\rho_{POCD}) - \frac{\partial(w_g \rho_{POCD})}{\partial z} - D_{POCD} + D_{h,POCL}. \tag{6}$$

where $\text{Dif}_A(\bullet) \equiv \frac{1}{A}\frac{\partial}{\partial z}\left(A k_s \frac{\partial \bullet}{\partial z}\right)$, $\text{Dif}(\bullet) \equiv \frac{\partial}{\partial z}\left(k_s \frac{\partial \bullet}{\partial z}\right)$ are diffusion operators, $w_g$ is a sedimentation velocity of POCD particles. Equations (4)-(6) do not contain $A$, because they are not derived by horizontal averaging, but follow from assumption of horizontal homogeneity of respective biogeochemical variable. This is caused by uncertainty of estimating the flux of these substances at the sloping interface between water and sediments. The r.h.s of these equations represent diffusion (assuming $k_s = k_{s,t} + k_{s,m}$ with the same eddy diffusivity $k_{s,t}$ and molecular diffusivity $k_{s,m}$ for all species; molecular dissusivity is not included in POCL and POCD equations), sources and sinks due to the following processes:

- dissolution/exsolution of gases at the bubble-water interface ($B_{CH_4}$, $B_{O_2}$ and $B_{CO2}$);

- photosynthesis ($P_{O_2}$, $P_{CO_2}$, $P_{POCL}$);

- respiration ($R_{O_2}, R_{CO_2}$, $R_{POCL}$);

- biochemical oxygen demand in the water column ($D_{O_2}$, $D_{CO_2}$, $D_{DOC}$, $D_{POCD}$);

- sedimentary oxygen demand ($S_{O_2}$, $S_{CO_2}$);

- methane aerobic oxidation in the water column ($O_{CH_4}$, $O_{O_2}$, $O_{CO_2}$);

- death of living species ($D_{h,POCL}$)

All variables in the above list are positive definite, excepting $B_{CH_4}, B_{O_2}$ and $B_{CO2}$ that may be either positive or negative. All concentrations in (1)-(3) are expressed in mol/m$^3$ that allows for simple relations of sinks and sources in different equations based on stoichiometry of the respective reactions. Organic carbon variables DOC, POCL and POCD in (4)-(6) are molar concentrations of carbon atoms contained in these organic groups. Terms $B_{CO_2}, P_{CO_2}, R_{CO_2}, D_{CO_2}, S_{CO_2}, O_{CO_2}$ in (3) possess "CO$_2$" subscript because carbon atoms are supplied to or removed from DIC of a solution in a form of CO$_2$.

In the following, the parameterizations of processes related to O$_2$ and CO$_2$ dynamics are described, whereas formulations for CH$_4$ processes are presented in (Stepanenko et al., 2016).

The formulations for photosynthesis, respiration, biochemical oxygen demand and sedimentary oxygen demand basically adopted from (Stefan and Fang, 1994) and (Hanson et al., 2004).

**1.2 Carbonate equilibrium**

Carbonate equilibrium means the equilibrium in the following reactions:

$$CO_2 + H_2O \leftrightarrow H^+ + HCO_3^-, \tag{7}$$
$$HCO_3^- \leftrightarrow H^+ + CO_3^{2-}. \tag{8}$$

Involving kinetic constants of these reactions yields, that the DIC reads

$$C_{DIC} \equiv C_{CO_2} + C_{HCO_3^-} + C_{CO_3^{2-}} = C_{CO_2} \left[1 + k_1 10^{pH} + k_1 k_2 10^{2pH}\right]. \tag{9}$$

Here, the constants are given by Arrhenius equation:

$$k_i = k_{i0} \exp\left[-\frac{E_{act,i}}{R}\left(\frac{1}{T} - \frac{1}{T_0}\right)\right], \ i = 1, 2, \tag{10}$$

$R$ – universal gas constant, $k_1 = 4.3*10^{-7}$ mol/l, $k_2 = 4.7*10^{-11}$ mol/l, $E_{act,1} = 7.66 * 10^3$ J/mol, $E_{act,2} = 1.49 * 10^4$ J/mol. Thus, $C_{CO_2}$ is readily calculated given $C_{DIC}$ value, and vice versa, where $pH$ is an external parameter.

Carbon atoms are added or removed from carbonate equilibrium system in a form of $CO_2$ during respiration, photosynthesis and organic chemical

and physical processes, hence the change of $C_{DIC}$ equals to number of $CO_2$ consumed or produced. This explains the sense of terms in equation (3). For obtaining $CO_2$ flux across bubble surface or $CO_2$ diffusive flux to the atmosphere, $C_{CO_2}$ is needed and is calculated from (9).

**1.3 Boundary conditions for dissolved gases in a water column**

The top boundary condition (at the lake-atmosphere interface) for any dissolved gas concentration in the case of open water has the form:

$$\frac{k_s}{h}\frac{\partial C}{\partial \xi}\bigg|_{\xi=0} = F_C, \tag{11}$$

where $C$ is $C_{CH_4}, C_{O_2}$ or $C_{CO_2}$, and $F_C$ is the diffusive flux of a gas into the atmosphere, positive upwards. This flux is calculated according to the widely used parameterization:

$$F_C = k_{ge}(C|_{\xi=0} - C_{ae}), \tag{12}$$

with $C_{ae}$ being the concentration of the gas in water equilibrated with the atmospheric concentration and described by Henry law and $k_{ge}$, m/s, denoting the gas exchange coefficient, the so-called "piston velocity". The latter is written as:

$$k_{ge} = k_{600}\sqrt{\frac{600}{Sc(T)}}, \tag{13}$$

with the Schmidt number $Sc(T)$ having individual values for different gases and being temperature-dependent. The $k_{600}$ coefficient has been a subject of numerous studies, and a number concepts have been put forward to quantify it (Donelan and Wanninkhof, 2002). The proper computation of this coefficient should account for the effects of a number of factors such as turbulence in adjacent layers of water and air, wave development and breaking, cool skin dynamics. The surface renewal model (MacIntyre et al., 2010; Heiskanen et al., 2014), used in LAKE2.0 model, "integrates" those effects through the near-surface dissipation rate of turbulent kinetic energy:

$$k_{600} = \frac{C_{1,SR}(\epsilon|_{\xi=0}\nu_w)^{\frac{1}{4}}}{\sqrt{600}}, \tag{14}$$

where $\nu_w$ designates molecular viscosity of water, $C_{1,SR} = 0.5$ is an empirical constant. TKE dissipation rate is available directly from $k - \epsilon$ closure.

When a lake is covered by ice, $F_C = 0$, which neglects contribution of diffusion through ice cracks.

**1.4 Photosynthesis**

The intensity of photosynthesis in terms of oxygen molecules production is expressed as:

$$P_{O_2} = \frac{P_{max} L_{min} \rho_{Chl-a}}{H_{sec} \mu_{O_2}}. \tag{15}$$

The denominator here serves co convert units in the r.h.s. from mg/(l*h) to mol/(m$^3$ s). The $P_{max}$ value expresses limitation of oxygen production by temperature in a form:

$$P_{max} = C_P \theta_P^{(T-T_0)}, \tag{16}$$

so that $C_P$ is a value of $P_{max}$ at the reference temperature $T = T_0$. The limitation of oxygen production by the available photosynthetically active radiation PAR ($S_{PAR}$) is given by the Haldane kinetics:

$$L_{min} = \frac{S_{PAR}(1 + 2\sqrt{C_{Lmin,1}/C_{Lmin,2}})}{S_{PAR} + C_{Lmin,1} + S_{PAR}^2/C_{Lmin,2}}. \tag{17}$$

The PAR intensity delivering maximum to a limiter $L_{min}$ (=1) is $S_{PAR} = \sqrt{C_{Lmin,1}C_{Lmin,2}}$. In the model, these coefficients are specified as (Stefan and Fang, 1994; Megard et al., 1984):

$$C_{Lmin,1} = C_{PAR}\theta_{PAR}^{(T-T_0)}, \tag{18}$$

$$C_{Lmin,2} = \mathrm{H}(T - T_{00})C_{Lmin,2,>T_{00}} + [1 - \mathrm{H}(T - T_{00})]C_{Lmin,2,<T_{00}}, \tag{19}$$

with H($\bullet$) denoting a Heavyside function, and $T_{00}$ standing for another reference temperature. It is seen from (17), that $L_{min} \to 0$ if $S_{PAR} \to 0$ and $S_{PAR} \to \infty$, i.e. PAR ihnibits photosynthesis at both low and high values of its intensity. The PAR instensity $S_{PAR}$ is expressed in a number of photons per square meter per hour, so that:

$$S_{PAR} = H_{sec} T_{J \to Eins} S_{PAR}^*, \tag{20}$$

where $H_{sec} = 3600$ s and $S^*_{PAR}$ is PAR intensity in W/m$^2$. The coefficient transforming from J to Einstein (Einstein is an energy of Avogadro number of photons), $T_{J \rightarrow Eins}$, is estimated assuming the uniform distribution of energy in PAR region, which yields:

$$T_{J \rightarrow Eins} = \frac{\lambda_{PAR}}{N_A h_P c}, \tag{21}$$

with $N_A, h_P, c$ denoting Avogadro number, Planck constant and the light speed in vacuum, respectively, all in SI units.

The treatment of chlorophyll-a concentration $\rho_{Chl-a}$ is given in Section 1.10.

Finally, from the gross photosynthesis reaction:

$$6CO_2 + 12H_2O + photons \rightarrow C_6H_{12}O_6 + 6O_2 + 6H_2O, \tag{22}$$

or, in a shortened form:

$$CO_2 + 2H_2O + photons \rightarrow CH_2O + O_2 + H_2O, \tag{23}$$

we see that the carbon dioxide consumption equals oxygen production, i.e. $P_{CO_2} = P_{O_2}$.

Equation (22) also implies that $P_{POCL} = P_{CO_2}$.

**1.5 Respiration**

P.Hanson et al. (Hanson et al., 2004) assume, that respiration is performed by "living particles", i.e. POCL, only in epilimnion, and may be scaled by gross primary production (i.e., photosynthesis rate), $R_{POCL} = \alpha_{POCL} P_{POCL}$, $\alpha_{POCL} = 0.8$. In contrast, we assume that this process happens at all depths where enough oxygen *in situ* to be used in respiration is available, with the same scaling. Evidently,

$$R_{O_2} = R_{CO_2} = \alpha_{POCL} P_{POCL}. \tag{24}$$

**1.6 Biochemical oxygen demand (BOD)**

We treat biochemical oxygen demand as a consumption of oxygen during degradation of dead organic particles (POCD) $D_{POCD}$ and dissolved organic

Table 1: Constants in photosynthesis model

| Constant | Units | Value |
|---|---|---|
| $C_P$ | h$^{-1}$ | 9.6 |
| $\theta_P$ | n/d | 1.036 |
| $T_0$ | °C | 20 |
| $T_{00}$ | °C | 10 |
| $\mu_{O_2}$ | g/mol | 32 |
| $H_{sec}$ | s | 3600 |
| $C_{PAR}$ | Einstein/(m$^2$*h) | 0.687 |
| $\theta_{PAR}$ | n/d | 1.086 |
| $C_{Lmin,2,>T_{00}}$ | Einstein/(m$^2$*h) | 15. |
| $C_{Lmin,2,<T_{00}}$ | Einstein/(m$^2$*h) | 5. |
| $\lambda_{PAR}$ | m | $5.5 * 10^{-7}$ (550 nm) |

carbon (DOC) $D_{DOC}$, following (Hanson et al., 2004); they suggest that $D_{POCD} = \rho_{POCD}/\tau_{POCD}$, $D_{DOC} = \rho_{DOC}/\tau_{DOC}$ with time scales $\tau_{POCD} = 20D_{sec}$, $\tau_{DOC} = 200D_{sec}$ ($D_{sec}$ is a number of seconds in a day). Thus, the BOD rate is:

$$D_{O_2} = D_{CO_2} = \left( \frac{\rho_{POCD}}{\tau_{POCD}} + \frac{\rho_{DOC}}{\tau_{DOC}} \right). \tag{25}$$

**1.7 Sedimentary oxygen demand (SOD)**

The sedimentary oxygen demand appears as a sink in (2) and in essence is the contribution of the vertical flux of $O_2$ at the lake's bottom to the horizontally averaged oxygen concentration:

$$S_{O_2} = -\frac{F_{SOD}}{A} \frac{\partial A}{\partial z}. \tag{26}$$

Basing on the argument that SOD is controlled by both diffusion (governed by Fickian law) and biochemical consumption (described by Michaelis-Menthen kinetics), (Walker and Snodgrass, 1986) derive:

$$F_{SOD} = \mu_\beta \frac{C_{O_2}}{k_{O_2,SOD} + C_{O_2}} + k_c C_{O_2}, \tag{27}$$

where $\mu_\beta$ is proportional to organics oxidation potential rate in sediments, and $k_c$ is the mass transfer coefficient. Both are thought to be exponentially dependent on temperature:

$$\mu_\beta = \mu_{\beta,0}\theta_{\mu_\beta}^{T-T_{\mu_\beta}}, k_c = k_{c,0}\theta_{k_c}^{T-T_{k_c}}. \tag{28}$$

The stoichiometry of SOD is assumed to be close to that of BOD (**??**), therefore, $S_{CO_2} = S_{O_2}$. Additionally, the flux of $O_2$ due to SOD at the lake bottom, $F_{SOD}$, is used as the bottom (lake deepest point) boundary condition for the oxygen equation (2).

Table 2: Constants in sedimentary oxygen demand model

| Constant | Units | Value |
|---|---|---|
| $\theta_{\mu_\beta}$ | n/d | 1.085 |
| $\theta_{k_c}$ | n/d | 1.103 |
| $T_{\mu_\beta}$ | K | 25 |
| $T_{k_c}$ | K | 20 |
| $\mu_{\beta,0}$ | mol/(m$^2$*s) | $0.5/(\mu_{O_2}D_{sec})$, $[\mu_{O_2}] = g/mol$ |
| $k_{c,0}$ | m/s | $0.045/D_{sec}$ |

**1.8 Exudates and death rate of POCL**

Hanson et al. suggest exudation to be scaled with photosynthesis rate, $E_{POCL} = \beta_{POCL}P_{POCL}$, $\beta_{POCL} = 0.03$ and the death rate to be defined as $D_{h,POCL} = \frac{\rho_{POCL}}{\tau_{Dh}}$, where time scale $\tau_{Dh}$ ranges from $1.1D_{sec}$ in hypolimnion to $33D_{sec}$ in epilimnion.

**1.9 Sedimentation of organic particles**

In the current model version we use the Stokes sedimentation velocity below the mixed layer:

$$w_s = \frac{4}{3A}\frac{\Delta gd^2}{\nu_m}, \tag{29}$$

and the high-Reynolds-number limit of this variable

$$w_s = \sqrt{\frac{4}{3B}\Delta g d} \tag{30}$$

in the mixed layer. Here, $\Delta = \rho_p/\rho_{w0} - 1$, $\rho_p$ is a particle's density, and $d$ – its diameter, the typical values for constants may be chosen as $A = 30.0$, and $B = 1.25$ (Song et al., 2008), and the density of organic particles as $1.25\ g/cm^3$ (Avnimelech et al., 2001).

**1.10 Chlorophyll-a dynamics**

The chlorophyll-a dynamics in the model follows a simple scheme suggested in (Stefan and Fang, 1994), where chlorophyll-a density is calculated as:

$$\rho_{Chl-a} = \rho_{Chl-a,0}H(H_a - z), \tag{31}$$

where the active layer, $H_a$, is a maximum value between mixed-layer depth, $H_{ML}$, and the photic zone depth, $H_{PZ}$. The mixed-layer depth is defined as the depth of maximum Brunt-Väisälä frequency, and the photic zone depth is estimated as the depth at which the PAR irradiance drops to 10% of its surface value. The mean chlorophyll-a concentration in the active layer, $\rho_{Chl-a,0}$, is assigned according to a trophic status of the lake: $2 * 10^{-3}$ mg/l for oligotrophic lakes, $6 * 10^{-3}$ mg/l for mesotrophic lakes and $15 * 10^{-3}$ mg/l for eutrophic lakes. In turn, the trophic status is formally defined from the water turbidity. The Secchi disk values of 2 m and 3.5 m are used to distinguish between eutrophic and mesotrophic, mesotrophic and oligotrophic states, respectively. These thresholds are expressed in the model through light extinction coefficient values, $\alpha$, using Poole and Atkins formula (Poole and Atkins, 2009):

$$\alpha = \frac{k_{PA}}{z_{SD}}, \tag{32}$$

where $z_{SD}$ is the Secchi disk depth and $k_{PA} = 1.7$. The above chlorophyll-a scheme is identical to that of (Stefan and Fang, 1994), excepting for it does not take into account the annual cycle of $\rho_{Chl-a,0}$.

**2 Sensitivity tests**

[Figure]

Figure S1: ML water temperature, original and modified model results and their errors when compared to the observations.

[Figure]

Figure S2: DO concentration at different chlorofill-a concentration values.

[Figure]

Figure S3: $CO_2$ concentration at different chlorofill-a concentration values.

[Figure]

Figure S4: Dissolved $CO_2$ concentrations at different pH values.

[Figure]

Figure S5: ML temperature at different hypolimnetic diffusivity coefficient values.

---

## Author Response (AR2)

**Numerical study of the seasonal thermal and gas regimes of the large artificial reservoir in Western Europe using LAKE2.0 model**

AUTHORS' RESPONSES TO THE REFEREE #2 FINAL COMMENTS

Maksim Iakunin[1], Victor Stepanenko[2] , Rui Salgado[1] , Miguel Potes[1] , Alexandra Penha[3,4] , Maria Helena Novais[3,4] , and Gonçalo Rodrigues[1]

*miakunin@uevora.pt*

[1]Department of Physics, ICT, Institute of Earth Sciences, University of Évora, 7000 Évora, Portugal
[2]Lomonosov Moscow State University, GSP-1, 119234, Leninskie Gory, 1, bld. 4, Moscow, Russia
[3]Water Laboratory, University of Évora, P.I.T.E. Rua da Barba Rala No1, 7005-345 Évora, Portugal
[4]Institute of Earth Sciences — ICT, University of Évora, Rua Romão Ramalho 59, 7000-671 Évora, Portugal

**Contents**

**Introduction. Document structure**

This document contains authors' responses to the comments of the Anonymous Referee. The document structure is the following:

- Referee's comments are numbered and given in *italic font*. General, specific, and technical comments come separately.

- Authors' response follows the comment and starts after **"Response:"** with normal font.

- The text from the article itself (if some changes are done, and if it is reasonable to provide it) is typed with `typewriter font` and separated from the response with an extra blank line.

- *Technical comments and mistakes* are not numbered, and authors' response follows immediately.

Reviewed manuscript with all the corrections is given after all responses. It contains the changes and proposals of **two** Referees and was prepared using LaTeXdiff package for better understanding of what has been changed.

**Anonymous Referee #2**

*General comments*

*I put quite a bit of effort into reviewing this paper, and I am pleased to see how thoroughly and thoughtfully the authors have responded to my comments. Below I suggest a few more minor edits and suggested changes to the manuscript. After making these changes the paper should be accepted for publication. I will not need to review it again.*

**Response:**   We thank the Reviewer for the the time and efforts they put into work. The paper was re-edited and corresponding modifications were made according to suggested changes.

**Specific changes**

*Pg 2 line 14 - vital activity » services*
Corrected

*Pg 2 line 24 - change acidity to pH*
Corrected

*Pg 4 line 21 has been » was*
Corrected

*Pg 4 line 22 definitely » for the remainder of the study*
Corrected

*Pg 5 line 6 processes is » processes are*
Corrected

*Pg 6 line 23 below the mixed layer of the euphotic zone?*
Corrected: involving subsurface turbulent kinetic energy dissipation rate below the mixed layer of the euphotic zone, provided by the $k - \epsilon$ closure.

*Pg 6 line 33 demonstates a big » undergoes a large*
Corrected

*Pg 7 line 4 allowed to improve the results » led to improved results*
Corrected

*Pg 8 fig 2 from the description in the paper it seems that the black dashed lines are to make the end of thermal stratification, but they clearly seem to be place on the heat maps at time of distinct thermal stratification (significant vertical temperatgure gradients). Check to see if there an error in the placement of these lines.*
To determine the beginning and end of stratification period we used a criterion from Wetzel's "Limnology": temperature gradient should be higher than 1 degree per 1 metre depth. We put the dashed lines (stratification borders) at the places where such conditions occurs stably several days in a row. It may seems from the Fig. 2 that stratification ends too early in 2018, however, this just corresponds to the criteria mentioned above. In the end of September 2018 there still was a thermal gradient, but less that 1 degree/metre. This was re-checked once again.

*Pg 8 line 18 cooling » cools*
Corrected

*Pg 9 I would suggest that you change*
*As in the real ML the temperature is not exactly constant, measurements from the sensor at 0.5 m depth were chosen for the comparison.*
*Since the vertical gradient of the measured ML temperature is not exactly constant , measurements from the sensor at 0.5 m depth were chosen to represent the mixed layer temperature in figure 3 .*
Changed

*Pg 10 I find this a little confusing*
*FLake provides ML depth, shape factor for the thermocline curve, ML and bottom temperature. It seems like these cannot be independent. The shape factor must in some way be dependent on the temperatures, likewise ML temperature must be dependent of the ML shape factor determining the ML depth. I think you just need to add a little more describing how this works.*
That's right, in general, these parameters are not independent. However, we don't explain the principle of FLake calculations here but speak about how does the model represents the profile. Unlike the LAKE model, FLake's output provides those variables, which can be used to calculate the temperature profile backwards. The following changes were made in the text:
FLake outputs include ML depth temperature, shape factor for the thermocline curve, and temperature at the bottom. Using these values it is possible to retrieve a water temperature profile.

*Pg 11 line 17 as well as the Flake » as well as Flake or as well as the Flake model*
Corrected

*Pg 15 line 6 this » these*
Corrected

*Pg 17 line 1. I don't quite understand what you mean by does not show large residuals. Fig 8b seems to me to have much higher residuals that fig 8a*
That's true, however, we wanted to point out there that after almost six months the model showed realistic results, quite close to monthly average value of $CO_2$.

*Pg 17 line 5 delete - values of O2*
Corrected

*Pg 17 line 6 delete - (underestimation of O2 values) and winter minimum (high overestimation)*
Corrected

*Pg 17 line 7 Supposedly » probably*
Corrected

*Pg 17 line 8 for elimination this flaws » to improve model performance*
Corrected

*Pg 17 line 8 inside the lakes » of lakes*
Corrected